# Information-Theoretic Bayesian Optimization for Bilevel Optimization Problems

## Abstract

A bilevel optimization problem consists of two optimization problems nested as an upper- and a lower-level problem, in which the optimality of the lower-level problem defines a constraint for the upper-level problem. This paper considers Bayesian optimization (BO) for the case that both the upper- and lower-levels involve expensive black-box functions. Because of its nested structure, bilevel optimization has a complex problem definition and, compared with other standard extensions of BO such as multi-objective or constraint settings, it has not been widely studied. We propose an information-theoretic approach that considers the information gain of both the upper- and lower-optimal solutions and values. This enables us to define a unified criterion that measures the benefit for both level problems, simultaneously. Further, we also show a practical lower bound based approach to evaluating the information gain. We empirically demonstrate the effectiveness of our proposed method through several benchmark datasets.

## 1 Introduction

The bilevel optimization is a standard formulation for a decision making problem that has a hierarchical structure. It consists of two optimization problems nested as an upper- and a lower-level problem, in which the optimality of the lower-level problem defines a constraint for the upper-level problem. For example, in the computational materials design, a target property should be optimized under the constraint of the energy minimization. Bilevel optimization techniques is applicable to hierarchical decision makings in a variety of contexts such as inverse optimal control (Suryan et al., 2016), chemical reaction optimization (Abbassi et al., 2021), and shape optimization (Herskovits et al., 2000).

We particularly focus on the case both level problems are defined by expensive black-box functions. Most of BO studies for bilevel optimization consider applying BO only to the upper-level problem (e.g., Kieffer et al., 2017; Dogan & Prestwich, 2023) as pointed out by (Chew et al., 2025). On the other hand, for example, consider the case that the upper- and lower- objective functions are defined through simulators of a subject of interest. If these simulators consist of expensive computations (such as quantum-mechanical calculations), in this bilevel optimization, both level problems are expensive to evaluate. For example, the simulator-based optimization of a physical property of inorganic crystals under the stability constraint (energy minimization) can be formulated as this class of problems.

In existing studies in which the lower-level problem is not expensive, typically, under a selected query for the upper-level problem, repeated queries to the lower-level problem is required, and further, the gradient of the lower-level problem is often assumed (e.g., Fu et al., 2024). Islam et al. (2018) and Wang et al. (2021) consider BO in both levels, but repeated queries on lower-level is still required. These approaches are not fully suitable when both levels are expensive black-boxes in which the gradient is not available. On the other hand, recently a few methods without those limitations have also been studied. Ekmekcioglu et al. (2024) combine the Thompson sampling on the upper-level query and a knowledge gradient-based extension of multi-task BO on the lower-level, but the theoretical justification for the combination of these two different criteria has not been revealed. Further, Chew et al. (2025) propose the well-known GP upper confidence bound (UCB) based approach to bilevel BO, called BILBO. Although BILBO has a theoretical regret guarantee,

in general, the performance of GP-UCB based methods depend on the selection of the balancing parameter of the exploitation and exploration, because the theoretically recommended value often does not provide the best performance (Srinivas et al., 2010).

We propose an information-theoretic approach that considers the simultaneous information gain for both the upper- and lower-optimal solutions and values, which we call bilevel information gain. This enables us to define a unified criterion that measures the benefit for both level problems simultaneously, which is not necessarily common in the case of bilevel methods as mentioned in the previous paragraph. Although the effectiveness of information-theoretic BO has been shown in several different contexts (e.g., Hennig & Schuler, 2012; Hernández-Lobato et al., 2014; Hoffman & Ghahramani, 2015; Wang & Jegelka, 2017; Hernández-Lobato et al., 2015; Hernandez-Lobato et al., 2016; Suzuki et al., 2020; Takeno et al., 2022a;b; Hvarfner et al., 2022; Tu et al., 2022), it has not been combined with bilevel optimization, to our knowledge. We first define bilevel information gain by extending the idea of the joint entropy search (Hvarfner et al., 2022; Tu et al., 2022). Unfortunately, the original definition of bilevel information gain is computationally intractable, and we show that a natural extension of the truncation based approximation, which has been widely employed in information-theoretic BO (e.g., Wang & Jegelka, 2017), can be derived. By combining the truncation based approximation and a variational lower bound (Takeno et al., 2022b), we obtain our criterion called Bilevel optimization via Lower-bound based Joint Entropy Search (BLJES). Further, while we mainly consider 'coupled setting' in which upper- and lower-level observations are obtained simultaneously, 'decoupled setting', in which a separate observation for each level is available, is also discussed. For example, in the case that the objective function values are outputs of some simulators (e.g., physical simulation), if a common simulator provides both level observations simultaneously, coupled setting is suitable, while if the upper- and the lower-level observations are from different simulators, decoupled setting can be more appropriate. We further propose an extension for the case that each level problem has inequality constraints (i.e., each level problem is a constraint problem).

Our contributions are summarized as follows.

- We show an information-theoretic formulation of bilevel BO, which has never been explored, to our knowledge. Bilevel information gain is defined to measure the benefit for both level problems.
- We derive a lower bound based approximation of bilevel information gain. We extend the standard truncation based approach in the single-level information-theoretic BO to the bilevel problem.
- We further propose extensions for decoupled setting and constraint problems. We show that our framework can handle these settings by a natural extension of bilevel information gain.

We demonstrate effectiveness of BLJES through functions generated from Gaussian processes and several benchmark problems.

## 2 Preliminaries

**Bilevel optimization.** Let $f : \mathcal{X} \times \Theta \to \mathbb{R}$ and $g : \mathcal{X} \times \Theta \to \mathbb{R}$ denote the upper- and the lower-level objective functions, respectively, both of which are assumed to be costly black-box functions. The upper- and the lower-level variables are denoted by $\boldsymbol{x} \in \mathcal{X}$ and $\boldsymbol{\theta} \in \Theta$, respectively, where $\mathcal{X} \subset \mathbb{R}^{d_\mathcal{X}}$ and $\Theta \subset \mathbb{R}^{d_\Theta}$. The bilevel optimization problem is formulated as:

$$
\begin{aligned}
\max_{\boldsymbol{x} \in \mathcal{X}} \ & f(\boldsymbol{x}, \boldsymbol{\theta}^*(\boldsymbol{x})) \\
\text{s.t. } & \boldsymbol{\theta}^*(\boldsymbol{x}) = \arg\max_{\boldsymbol{\theta} \in \Theta} g(\boldsymbol{x}, \boldsymbol{\theta}),
\end{aligned}
\tag{1}
$$

where $\boldsymbol{\theta}^*(\boldsymbol{x})$ represents the optimal solution of the lower-level problem for a given upper-level variable $\boldsymbol{x}$. For simplicity, we assume the lower-level optimum $\boldsymbol{\theta}^*(\boldsymbol{x})$ is uniquely determined for each $\boldsymbol{x}$ (called optimistic setting (Sinha et al., 2020)). The bilevel optimal solution is denoted by $(\boldsymbol{x}^*, \boldsymbol{\theta}^*)$, while the lower-level optimum corresponding to a given $\boldsymbol{x}$ is written as $(\boldsymbol{x}, \boldsymbol{\theta}^*(\boldsymbol{x}))$, noting that $\boldsymbol{\theta}^* = \boldsymbol{\theta}^*(\boldsymbol{x}^*)$. The upper- and the lower-level optimal values are denoted by $f^* := f(\boldsymbol{x}^*, \boldsymbol{\theta}^*)$ and $g^* := g(\boldsymbol{x}^*, \boldsymbol{\theta}^*)$, respectively. Figure 1 shows an illustration. Observations of the

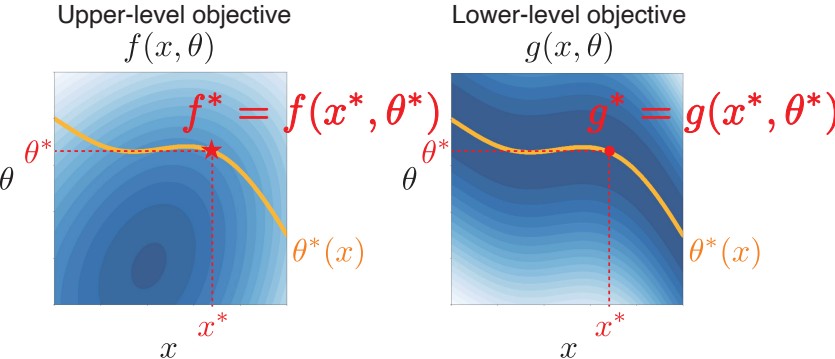

Figure 1: Example of Bilevel optimization ($d_\chi = 1, d_\Theta = 1$) and its optimal solution. For each upper-level variable $x$, the feasible solution is defined by $\theta^*(x) = \arg\max_\theta g(x, \theta)$, shown as the orange line. The optimal solution $(x^*, \theta^*)$ is the maximizer of $f$ on the orange line.

objective functions contain additive Gaussian noise $y^f_{(x,\theta)} := f(x, \theta) + \epsilon^f$, $\epsilon^f \sim \mathcal{N}(0, \{\sigma^f_{\text{noise}}\}^2)$, and $y^g_{(x,\theta)} := g(x, \theta) + \epsilon^g$, $\epsilon^g \sim \mathcal{N}(0, \{\sigma^g_{\text{noise}}\}^2)$, where $\sigma^f_{\text{noise}}$ and $\sigma^g_{\text{noise}}$ are the noise standard deviations, respectively. Let $\mathcal{D}_t := \{(x_i, \theta_i, y^f_i, y^g_i)\}^n_{i=1}$ be the dataset that we have at the $t$-th iteration of BO, where $y^f_i := y^f_{(x_i, \theta_i)}$ and $y^g_i := y^g_{(x_i, \theta_i)}$. The number of observed points $n$ is $t + n_0$, where $n_0$ is the number of the initial observations.

**Gaussian process.** The upper- and the lower-level objective functions are each modeled by independent Gaussian processes (GPs) with kernel functions $k^f((x, \theta), (x', \theta'))$ and $k^g((x, \theta), (x', \theta'))$, respectively. Given the dataset $\mathcal{D}_t$, the predictive distribution of an objective function $h \in \{f, g\}$ at a point $(x, \theta)$ is expressed as:

$$h(x, \theta) \mid \mathcal{D}_t \sim \mathcal{N}(\mu^h_t(x, \theta), \{\sigma^h_t(x, \theta)\}^2), \tag{2}$$
$$\mu^h_t(x, \theta) = k^{h\top}(K^h + \{\sigma^h_{\text{noise}}\}^2 I)^{-1} y^h,$$
$$\{\sigma^h_t(x, \theta)\}^2 = k^h((x, \theta), (x, \theta)) - k^{h\top}(K^h + \{\sigma^h_{\text{noise}}\}^2 I)^{-1} k^h,$$

where $y^h = (y^h_1, \ldots, y^h_n)^\top$, $k^h = (k^h((x, \theta), (x_1, \theta_1)), \ldots, k^h((x, \theta), (x_t, \theta_n)))^\top$, and $K^h \in \mathbb{R}^{n \times n}$ is the kernel matrix with an entry $k^h((x_i, \theta_i), (x_j, \theta_j))$ at a position $(i, j)$. Here, $I \in \mathbb{R}^{n \times n}$ denotes the identity matrix.

**Bayesian optimization.** We consider BO for the bilevel optimization problem (1). Bayesian optimization is a method for efficiently optimizing black-box functions with a limited number of samples. At step $t$, GPs are fitted to the dataset $\mathcal{D}_t$, and the next query point is determined as $\arg\max_{x,\theta} \alpha_t(x, \theta)$, where $\alpha_t(x, \theta)$ denotes the acquisition function. After sampling the query point, the newly obtained data are added to the dataset, and the GPs are refitted.

## 3 Bilevel Optimization via Lower-Bound based Joint Entropy Search

We consider bilevel BO based on the information gain for the optimal solutions and values ($x^*, \theta^*, f^*$, and $g^*$) achieved by next observations $y^f_{(x,\theta)}$ and $y^g_{(x,\theta)}$, which we call bilevel information gain. Note that we regard the optimal $(x^*, \theta^*, f^*, g^*)$ as random variables defined by the predictive distributions of the objective functions $f(x, \theta)$ and $g(x, \theta)$. Our approach combines the concept of entropy search (Hennig & Schuler, 2012), in particular, joint entropy search (Hvarfner et al., 2022; Tu et al., 2022), and a variational lower bound based approximation of mutual information (MI). We refer to our proposed method as *Bilevel optimization via Lower-bound based Joint Entropy Search* (BLJES).

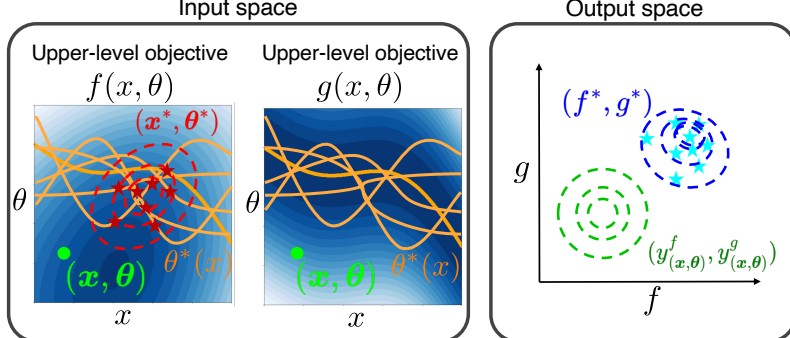

Figure 2: Schematic illustration of variables in $\text{MI}(y^f_{(\boldsymbol{x},\boldsymbol{\theta})}, y^g_{(\boldsymbol{x},\boldsymbol{\theta})} \; ; \; f^*, g^*, \boldsymbol{x}^*, \boldsymbol{\theta}^* \mid \mathcal{D}_t)$. The predictions at current $(\boldsymbol{x}, \boldsymbol{\theta})$ are $y^f_{(\boldsymbol{x},\boldsymbol{\theta})}$ and $y^g_{(\boldsymbol{x},\boldsymbol{\theta})}$, depicted as the green distribution in the output space. At the optimal solution $(\boldsymbol{x}^*, \boldsymbol{\theta}^*)$, the objective function values are $f^* = f(\boldsymbol{x}^*, \boldsymbol{\theta}^*)$ and $g^* = g(\boldsymbol{x}^*, \boldsymbol{\theta}^*)$. The optimal values $f^*$ and $g^*$ are represented as the blue distribution in the output space. The optimal solution $(\boldsymbol{x}^*, \boldsymbol{\theta}^*)$, represented as the red distribution in the input space, should exist on $\boldsymbol{\theta}^*(\boldsymbol{x})$, i.e., the optimal point of the lower-level problem. The red and blue stars are 'samples' of the optimal solutions $(\boldsymbol{x}^*, \boldsymbol{\theta}^*)$ and values $(f^*, g^*)$, respectively.

### 3.1 Lower Bound of Mutual Information

Bilevel information gain is represented as the MI between the candidate observations $(y^f_{(\boldsymbol{x},\boldsymbol{\theta})}, y^g_{(\boldsymbol{x},\boldsymbol{\theta})})$ and the set of the optimal solutions and their upper- and lower-objective values $\{f^*, g^*, \boldsymbol{x}^*, \boldsymbol{\theta}^*\}$:

$$\text{MI}(y^f_{(\boldsymbol{x},\boldsymbol{\theta})}, y^g_{(\boldsymbol{x},\boldsymbol{\theta})} \; ; \; f^*, g^*, \boldsymbol{x}^*, \boldsymbol{\theta}^* \mid \mathcal{D}_t),$$

for which an illustration is shown in Fig. 2. This criterion naturally allows simultaneous consideration of both the upper- and the lower-objectives. Since the direct evaluation of this MI is difficult, we employ a lower bound based approximation. Let $\Omega \coloneqq \{y^f_{(\boldsymbol{x},\boldsymbol{\theta})}, y^g_{(\boldsymbol{x},\boldsymbol{\theta})}, f^*, g^*, \boldsymbol{x}^*, \boldsymbol{\theta}^*\}$. Our lower bound of the MI is derived by a technique that is often used in the context of the variational approximation (e.g., Poole et al., 2019):

$$\text{MI}(y^f_{(\boldsymbol{x},\boldsymbol{\theta})}, y^g_{(\boldsymbol{x},\boldsymbol{\theta})} \; ; \; f^*, g^*, \boldsymbol{x}^*, \boldsymbol{\theta}^* \mid \mathcal{D}_t) = \mathbb{E}_\Omega \left[ \log \frac{p(y^f_{(\boldsymbol{x},\boldsymbol{\theta})}, y^g_{(\boldsymbol{x},\boldsymbol{\theta})} \mid f^*, g^*, \boldsymbol{x}^*, \boldsymbol{\theta}^*, \mathcal{D}_t)}{p(y^f_{(\boldsymbol{x},\boldsymbol{\theta})}, y^g_{(\boldsymbol{x},\boldsymbol{\theta})} \mid \mathcal{D}_t)} \right]$$

$$= \mathbb{E}_{f^*, g^*, \boldsymbol{x}^*, \boldsymbol{\theta}^*} \left[ \mathbb{E}_{y^f_{(\boldsymbol{x},\boldsymbol{\theta})}, y^g_{(\boldsymbol{x},\boldsymbol{\theta})} \mid f^*, g^*, \boldsymbol{x}^*, \boldsymbol{\theta}^*, \mathcal{D}_t} \left[ \log \frac{q(y^f_{(\boldsymbol{x},\boldsymbol{\theta})}, y^g_{(\boldsymbol{x},\boldsymbol{\theta})} \mid f^*, g^*, \boldsymbol{x}^*, \boldsymbol{\theta}^*, \mathcal{D}_t)}{p(y^f_{(\boldsymbol{x},\boldsymbol{\theta})}, y^g_{(\boldsymbol{x},\boldsymbol{\theta})} \mid \mathcal{D}_t)} \right] \right.$$

$$\left. + \text{KL} \left( p(y^f_{(\boldsymbol{x},\boldsymbol{\theta})}, y^g_{(\boldsymbol{x},\boldsymbol{\theta})} \mid f^*, g^*, \boldsymbol{x}^*, \boldsymbol{\theta}^*, \mathcal{D}_t) \parallel q(y^f_{(\boldsymbol{x},\boldsymbol{\theta})}, y^g_{(\boldsymbol{x},\boldsymbol{\theta})} \mid f^*, g^*, \boldsymbol{x}^*, \boldsymbol{\theta}^*, \mathcal{D}_t) \right) \right]$$

$$\geq \mathbb{E}_\Omega \left[ \log \frac{q(y^f_{(\boldsymbol{x},\boldsymbol{\theta})}, y^g_{(\boldsymbol{x},\boldsymbol{\theta})} \mid f^*, g^*, \boldsymbol{x}^*, \boldsymbol{\theta}^*, \mathcal{D}_t)}{p(y^f_{(\boldsymbol{x},\boldsymbol{\theta})}, y^g_{(\boldsymbol{x},\boldsymbol{\theta})} \mid \mathcal{D}_t)} \right] =: \text{LB}(\boldsymbol{x}, \boldsymbol{\theta}), \quad (3)$$

where KL is Kullback-Leibler (KL) divergence and $q(y^f_{(\boldsymbol{x},\boldsymbol{\theta})}, y^g_{(\boldsymbol{x},\boldsymbol{\theta})} \mid f^*, g^*, \boldsymbol{x}^*, \boldsymbol{\theta}^*, \mathcal{D}_t)$ is a variational distribution ($q$ can be any density function as far as the KL divergence can be defined). The inequality of the last line can be taken because the KL divergence is non-negative (the equality holds when $p(y^f_{(\boldsymbol{x},\boldsymbol{\theta})}, y^g_{(\boldsymbol{x},\boldsymbol{\theta})} \mid f^*, g^*, \boldsymbol{x}^*, \boldsymbol{\theta}^*, \mathcal{D}_t) = q(y^f_{(\boldsymbol{x},\boldsymbol{\theta})}, y^g_{(\boldsymbol{x},\boldsymbol{\theta})} \mid f^*, g^*, \boldsymbol{x}^*, \boldsymbol{\theta}^*, \mathcal{D}_t)$). Similar lower bounds of the MI have been used in information-theoretic multi-objective and constraint BO (Ishikura & Karasuyama, 2025; Takeno et al., 2022b).

The variational distribution $q$ is an approximation of $p(y^f_{(\boldsymbol{x},\boldsymbol{\theta})}, y^g_{(\boldsymbol{x},\boldsymbol{\theta})} \mid f^*, g^*, \boldsymbol{x}^*, \boldsymbol{\theta}^*, \mathcal{D}_t)$ for which an exact analytical representation is difficult to know. The difficulty is in the conditioning by the optimal solutions and values, for which the most widely accepted approach in information-theoretic BO is

to use truncated distributions (e.g., Wang & Jegelka, 2017; Suzuki et al., 2020; Tu et al., 2022). For example, in the case of well-known max-value entropy search (MES), proposed by (Wang & Jegelka, 2017) for the standard single-level problem $\max_x f(x)$, the predictive distribution conditioning on the max-value $f^*_{\text{unc}} := \arg\max_x f(x)$ is approximated by the truncated normal distribution, i.e., $p(f(x) \mid f^*_{\text{unc}}) \approx p(f(x) \mid f(x) \leq f^*_{\text{unc}})$. When $f^*_{\text{unc}}$ is given, $f(x') \leq f^*_{\text{unc}}$ should hold for any $x'$ (and there should exist at least one $x'$ such that $f(x') = f^*_{\text{unc}}$), while MES simplifies this condition so that $f(x) \leq f^*_{\text{unc}}$ holds only for the current $x$. Similar simplifications have been employed by most of information-theoretic BO algorithms and shown superior performance.

We extend the truncation based approach to our bilevel problem as follows.

$$q(y^f_{(x,\theta)}, y^g_{(x,\theta)} \mid f^*, g^*, x^*, \theta^*, \mathcal{D}_t) := p(y^f_{(x,\theta)}, y^g_{(x,\theta)} \mid f(x, \theta^*(x)) \leq f^*, g(x^*, \theta) \leq g^*, \mathcal{D}_t^+),$$
(4)

where $\mathcal{D}_t^+ = \mathcal{D}_t \cup \{(x^*, \theta^*, f^*, g^*)\}$ is the dataset augmented by the optimal point $(x^*, \theta^*, f^*, g^*)$. The right hand side has the three conditions, each of which can be interpreted as follows.

- When $f^*$ is given, $f(x', \theta^*(x')) \leq f^*$ should hold for $\forall x'$. However, this condition is computationally intractable as mentioned for the case of MES. Based on a similar idea of MES, the condition $f(x, \theta^*(x)) \leq f^*$ is only imposed on the current $x$.
- When $g^*$ is given, $g(x^*, \theta') \leq g^*$ should hold for $\forall \theta'$. We replace it with $g(x^*, \theta) \leq g^*$ in which the inequality is only imposed on the current $\theta$.
- In the right hand side of (4), $\mathcal{D}_t$ is replaced $\mathcal{D}_t^+$. This condition can impose that the GPs satisfy $f(x^*, \theta^*) = f^*$ and $g(x^*, \theta^*) = g^*$ by adding $(x^*, \theta^*, f^*, g^*)$ into the training data.

By substituting (4) into (3) and using the conditional independence of $y^f_{(x,\theta)}$ and $y^g_{(x,\theta)}$ in the right hand side of (4), we see

$$\text{LB}(x, \theta) = \mathbb{E}_\Omega \left[ \log \frac{p(y^f_{(x,\theta)}, y^g_{(x,\theta)} \mid f(x, \theta^*(x)) \leq f^*, g(x^*, \theta) \leq g^*, \mathcal{D}_t^+)}{p(y^f_{(x,\theta)}, y^g_{(x,\theta)} \mid \mathcal{D}_t)} \right]$$

$$= \mathbb{E}_\Omega \left[ \log \frac{p(y^f_{(x,\theta)} \mid f(x, \theta^*(x)) \leq f^*, \mathcal{D}_t^+)}{p(y^f_{(x,\theta)} \mid \mathcal{D}_t)} + \log \frac{p(y^g_{(x,\theta)} \mid g(x^*, \theta) \leq g^*, \mathcal{D}_t^+)}{p(y^g_{(x,\theta)} \mid \mathcal{D}_t)} \right]. \quad (5)$$

The inside of the expectation (5) can be analytically derived. For both the first and second terms, the denominators are the predictive distribution of the GPs, whose density can be obtained from (2). Next, we consider the numerator of the first term of (5). Although the truncation is imposed on $f(x, \theta^*(x))$, the distribution is for $y^f_{(x,\theta)}$ that has different input point $(x, \theta)$ from the truncated point, unlike the case of MES. The following theorem shows that the analytical representation can still be derived even with this difference:

**Theorem 3.1.** *Let* $(m^f_1, s^f_1)$, $(m^f_2, s^f_2)$, *and* $(m^f_3, s^f_3)$ *be the mean and standard deviation of* $p(f(x, \theta^*(x)) \mid y^f_{(x,\theta)}, \mathcal{D}_t^+)$, $p(f(x, \theta^*(x)) \mid \mathcal{D}_t^+)$, *and* $p(y^f_{(x,\theta)} \mid \mathcal{D}_t^+)$, *respectively. Then,*

$$p(y^f_{(x,\theta)} \mid f(x, \theta^*(x)) \leq f^*, \mathcal{D}_t^+) = \begin{cases} \Phi\left(\frac{f^* - m^f_1}{s^f_1}\right) \phi\left(\frac{y^f_{(x,\theta)} - m^f_3}{s^f_3}\right) \bigg/ \left\{ \Phi\left(\frac{f^* - m^f_2}{s^f_2}\right) s^f_3 \right\} & \text{if } x \neq x^*, \\ \phi\left(\frac{y^f_{(x,\theta)} - m^f_3}{s^f_3}\right) \bigg/ s^f_3 & \text{otherwise,} \end{cases}$$
(6)

*where $\phi$ and $\Phi$ are the probability density function (PDF) and cumulative density function (CDF) of the standard normal distribution, respectively.*

The proof is in Appendix A.1. Note that all of $\{(m^f_i, s^f_i)\}_{i=1}^3$ can be analytically calculated from the GP posterior of $f$ for which details are also show in Appendix A.1. For the numerator in the second term of (5) can be reduced to the similar analytical form, which is shown in Appendix A.2. As a result, we obtain an analytical form of the inside of the expectation (5). We also show that our lower bound (5) can be derived through an independence assumption on the posterior, described in Appendix B.

## 3.2 COMPUTATIONS

The expectation of (5) is approximated by the Monte-Carlo method by sampling $\Omega :=$ $\{y^f_{(x,\theta)}, y^g_{(x,\theta)}, f^*, g^*, x^*, \theta^*\}$. The sample of all the elements of $\Omega$ can be obtained through the sample of the objective functions $f$ and $g$. We use random Fourier feature (RFF) (Rahimi & Recht, 2008), by which the GP posterior can be approximated by the Bayesian linear model. For a $D$-dimensional RFF vector $\phi(x, \theta) \in \mathbb{R}^D$, the linear model $w^{h\top}\phi(x, \theta)$ can be constructed, where $w^h \in \mathbb{R}^D$ is a parameter vector and $h \in \{f, g\}$ represents one of objective functions. By sampling $w^h$ from the posterior, approximate sample paths of $f$ and $g$ are obtained, which denoted as $\tilde{f}$ and $\tilde{g}$, respectively. Then, the sample of $(f^*, g^*, x^*, \theta^*)$ is obtained through $\max_x \tilde{f}(x, \tilde{\theta}^*(x))$ s.t. $\tilde{\theta}^*(x) = \arg\max_\theta \tilde{g}(x, \theta)$, which can be seen as a white-box bilevel optimization problem. Since both $\tilde{f}$ and $\tilde{g}$ are represented by the Bayesian liner model, they are differentiable. Then, the gradient $\partial \tilde{f}(x, \theta^*(x))/\partial x$ can be obtained through the implicit function theorem (see Appendix C for detail), by which standard gradient based optimization methods can be applied. The samples of $y^f_{(x,\theta)}$ and $y^g_{(x,\theta)}$ can be obtained by adding the random noise from $\mathcal{N}(0, \{\sigma^f_{\text{noise}}\}^2)$ and $\mathcal{N}(0, \{\sigma^g_{\text{noise}}\}^2)$ to the sampled $f(x, \theta)$ and $g(x, \theta)$, respectively.

Let $K$ be the number of samplings of $\Omega$. In a variety of contexts of information-theoretic BO (e.g., Wang & Jegelka, 2017), the superior performance has been repeatedly shown with small $K$ settings (e.g., 10). After obtaining $K$ samples of $\Omega$, the Monte-Carlo approximation of (5) can be calculated for any $(x, \theta)$ (Note that these $K$ samples are reused during the acquisition function optimization without regenerating for each candidate $(x, \theta)$). Since (5) contains $\theta^*(x)$, the maximization of the approximate (5) is also bilevel optimization, for which gradient-based methods can also be applied through the implicit function theorem (details are also in Appendix C).

## 4 EXTENSIONS

We here describe two extensions of BLJES, which are for decoupled setting and constraint problems.

## 4.1 DECOUPLED SETTING

We mainly consider the setting in which $y^f_{(x,\theta)}$ and $y^g_{(x,\theta)}$ are observed simultaneously, which we call 'coupled' setting. On the other hand, only one of $y^f_{(x,\theta)}$ or $y^g_{(x,\theta)}$ can be separately observed in some scenarios. In this paper, this setting is called 'decoupled' setting, inspired by the similar setting in multi-objective BO (Hernandez-Lobato et al., 2016).

A natural criterion for decoupled setting is information gain obtained by only one of $y^f_{(x,\theta)}$ or $y^g_{(x,\theta)}$, for which the lower bounds can be derived by the almost same way as (3):

$$\text{MI}(y^f_{(x,\theta)} \; ; \; f^*, g^*, x^*, \theta^* \mid \mathcal{D}_t) \geq \mathbb{E}_\Omega \left[ \log \frac{p(y^f_{(x,\theta)} \mid f(x, \theta^*(x)) \leq f^*, \mathcal{D}_t^+)}{p(y^f_{(x,\theta)} \mid \mathcal{D}_t)} \right], \quad (7)$$

$$\text{MI}(y^g_{(x,\theta)} \; ; \; f^*, g^*, x^*, \theta^* \mid \mathcal{D}_t) \geq \mathbb{E}_\Omega \left[ \log \frac{p(y^g_{(x,\theta)} \mid g(x^*, \theta) \leq g^*, \mathcal{D}_t^+)}{p(y^g_{(x,\theta)} \mid \mathcal{D}_t)} \right]. \quad (8)$$

The derivation is in Appendix D. For both the inside of the expectation of (7) and (8), the analytical calculations shown in section 3.1 can be used. The expectation is approximated by Monte-Carlo sampling of $\Omega$, which is also same as coupled setting. As a result, the decision making not only for selecting $(x, \theta)$, but also selecting the upper- or the lower-observation (i.e., $y^f_{(x,\theta)}$ or $y^g_{(x,\theta)}$ ) can be performed.

## 4.2 INCORPORATING CONSTRAINT PROBLEMS

In a more general formulation of bilevel optimization, constraints are imposed on both of the upper- and the lower-level problems. When we have $N$ and $M$ inequality constraints for the upper- and the

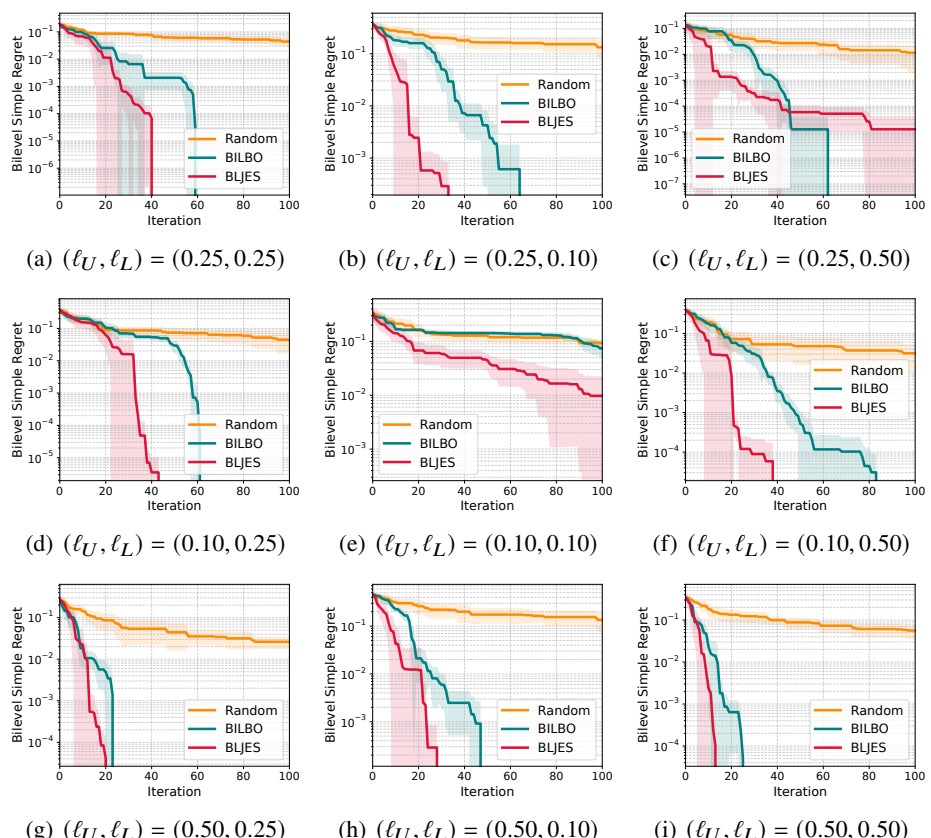

Figure 3: Regret comparison on functions from the GP prior.

lower-level problems, respectively, the bilevel optimization problem is written as

$$\max_{\boldsymbol{x} \in \mathcal{X}} f(\boldsymbol{x}, \boldsymbol{\theta}^*(\boldsymbol{x}))$$

$$\text{s.t. } c_n^U(\boldsymbol{x}, \boldsymbol{\theta}^*(\boldsymbol{x})) \geq 0, n = 1, \dots, N$$

$$\boldsymbol{\theta}^*(\boldsymbol{x}) = \arg\max_{\boldsymbol{\theta} \in \Theta} \{g(\boldsymbol{x}, \boldsymbol{\theta}) \mid c_m^L(\boldsymbol{x}, \boldsymbol{\theta}) \geq 0, m = 1, \dots, M\},$$

where $c^U : \mathbb{R}^{d_{\mathcal{X}}+d_\Theta} \to \mathbb{R}$ and $c^L : \mathbb{R}^{d_{\mathcal{X}}+d_\Theta} \to \mathbb{R}$ are constraint functions. We assume that $c^U$ and $c^L$ are also expensive black-box functions and modeled by the independent GPs.

For constraint BO, Takeno et al. (2022b) show an information-theoretic approach based on a lower bound with a truncated variational distribution. By combining the truncation shown by (Takeno et al., 2022b) and our bilevel information gain, we can extend BLJES to the bilevel constraint problem. In particular, the conditioning on the predictive distributions by the optimal points are required to extend. For example, if $f^*$ is given, the inequality $f(\boldsymbol{x}, \boldsymbol{\theta}^*(\boldsymbol{x})) \leq f^*$ is imposed only when the constraints $c_n^U(\boldsymbol{x}, \boldsymbol{\theta}^*(\boldsymbol{x})) \geq 0, n = 1, \dots, N$ hold (if the constraints are not satisfied, $f(\boldsymbol{x}, \boldsymbol{\theta}^*(\boldsymbol{x}))$ is not truncated). Details are in Appendix E.

## 5 EXPERIMENTS

We evaluated the performance of BLJES by using sample path functions from the GP prior and several benchmark functions. For baselines, we employed Random selection and BILBO. The initial number of observations was set $n_0 = 5$ random points. The both level observations contain an additive noise whose mean is 0 and standard deviation is $10^{-3}$. Each experiment was performed 10 times with different initial points. We used the Gaussian kernel for both the GPs of $f$ and $g$, in which the prior mean, the kernel length-scale, the output scale, and the noise variance are optimized

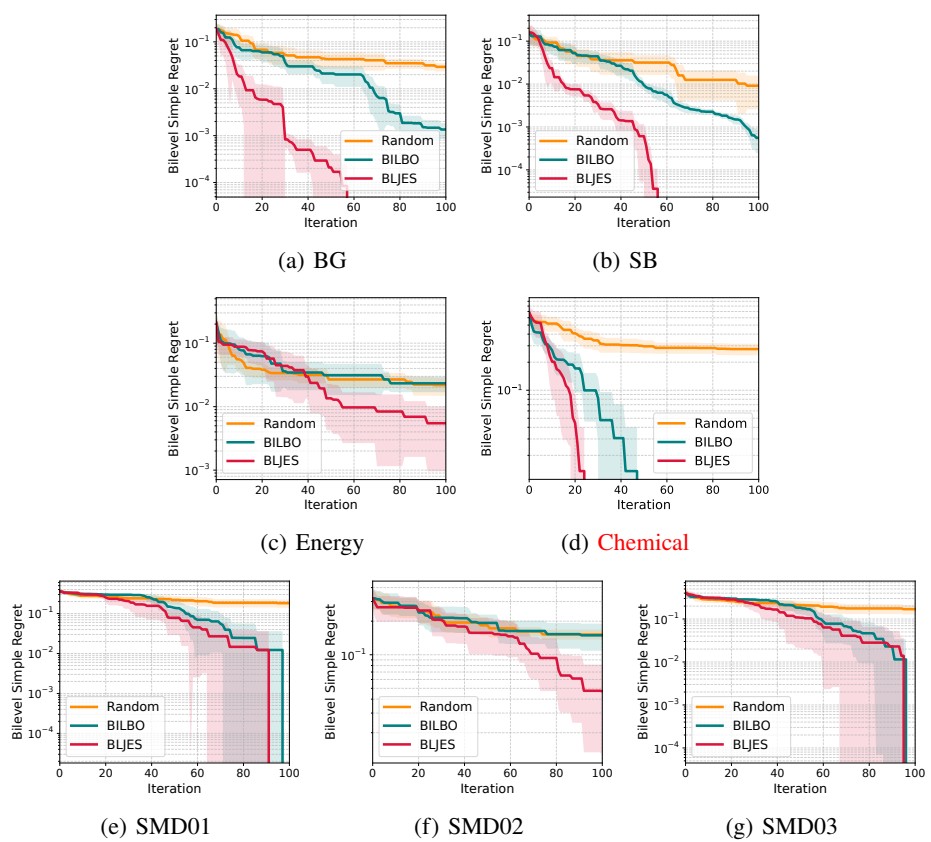

Figure 4: Regret comparison on benchmark problems.

by the marginal likelihood at every iteration. In BLJES, the number of Mont-Carlo samples was set as $K = 30$. We here employed the pool setting (query candidates are finite grid points defined later) because BILBO is proposed for the finite domain setting.

For the metric at the $t$-th iteration, we used the following criterion, denoted as bilevel simple regret:

$$\min_{i \in [n0+t]} \max_{h \in \{f,g\}} r_h(\boldsymbol{x}_i, \boldsymbol{\theta}_i), \tag{9}$$

where

$$r_f(\boldsymbol{x}_i, \boldsymbol{\theta}_i) = \max(0, f^* - f(\boldsymbol{x}_i, \boldsymbol{\theta}_i)) / (f^* - \min_{\boldsymbol{x}, \boldsymbol{\theta}} f(\boldsymbol{x}, \boldsymbol{\theta})),$$

$$r_g(\boldsymbol{x}_i, \boldsymbol{\theta}_i) = (g(\boldsymbol{x}_i, \boldsymbol{\theta}^*(\boldsymbol{x}_i)) - g(\boldsymbol{x}_i, \boldsymbol{\theta}_i)) / (g(\boldsymbol{x}_i, \boldsymbol{\theta}^*(\boldsymbol{x}_i)) - \min_{\boldsymbol{\theta}} g(\boldsymbol{x}_i, \boldsymbol{\theta})).$$

Our metric (9) takes the larger value between $r_f(\boldsymbol{x}_i, \boldsymbol{\theta}_i)$, which represents the regret of the upper-level problem, and $r_g(\boldsymbol{x}_i, \boldsymbol{\theta}_i)$, which represents those of the lower-level problem. Since $f(\boldsymbol{x}_i, \boldsymbol{\theta}_i)$ can be larger than $f^*$, the 'max' operation is taken to guarantee $r_f(\boldsymbol{x}_i, \boldsymbol{\theta}_i) \geq 0$, while the numerator of $r_g(\boldsymbol{x}_i, \boldsymbol{\theta}_i)$ is non-negative without 'max' from the definition of $\boldsymbol{\theta}^*(\boldsymbol{x}_i)$. The denominators of $r_f(\boldsymbol{x}_i, \boldsymbol{\theta}_i)$ and $r_g(\boldsymbol{x}_i, \boldsymbol{\theta}_i)$ are for absorbing the scale difference of two objectives. In (9), we employed the best value obtained during the entire search procedure by taking the minimum with respect to observed points.

We first provide the results on coupled setting for GP sample path functions (section 5.1) and benchmark functions (section 5.2). Further, the results on decoupled setting (section 5.3) and different $K$ settings (section 5.4) are also reported. Appendix presents other details of the settings (Appendix F.1) and results such as a larger noise setting (Appendix F.2), the continuous domain (Appendix F.5), constraint problems (Appendix F.6), higher dimensional settings (Appendix F.7), comparison with a simplified variant of MLJES (Appendix F.8), the effect of RFF approximation (Appendix F.9), and the combined error of the MC sampling and RFF (Appendix F.10)..

### 5.1 SAMPLE PATH FROM GP PRIOR

We first used the sample path from the GP prior as the true objective functions, i.e., $f \sim \mathcal{GP}(0, k)$ and $g \sim \mathcal{GP}(0, k)$, in which $k$ is the Gaussian kernel $k((\boldsymbol{x}, \boldsymbol{\theta}), (\boldsymbol{x}', \boldsymbol{\theta}')) = \exp\{-(\|\boldsymbol{x} - \boldsymbol{x}'\|^2 + \|\boldsymbol{\theta} - \boldsymbol{\theta}'\|^2)/(2\ell^2)\}$. For the length scale $\ell$, we use different values $\ell_U \in \{0.25, 0.10, 0.50\}$ for $f$ and $\ell_L \in \{0.25, 0.10, 0.50\}$ for $g$, respectively. The input space is $d_\mathcal{X} = d_\Theta = 1$ and $[0, 1]$ for each. The candidate points are a combination of 100 grid points in each dimension ($100^2$ points).

The results are shown in Fig. 3. Overall, BLJES shows superior performance for a variety of the length scale functions. Only for $(\ell_U, \ell_L) = (0.25, 0.50)$, BILBO decreased the regret to 0 faster at about 60 iterations, but BLJES also reached the small value ($10^{-4}$) around that iterations.

### 5.2 BENCHMARK FUNCTIONS

We here used six benchmark problems. Two functions are created by combining benchmark functions of single-level optimization. In the first problem, denoted as BG, the upper objective is BraninHoo ($d_\mathcal{X} = 1$) and the lower objective Goldstein-price ($d_\Theta = 1$), which was used in (Chew et al., 2025). In the second problem, denoted as SB, the upper objective is SixHumpCamel ($d_\mathcal{X} = 1$) and the lower objective BraninHoo ($d_\Theta = 1$), which was used in (Ekmekcioglu et al., 2024). The third problem, denoted as Energy, is a simulator based energy market problem ($d_\mathcal{X} = 2$ and $d_\Theta = 2$), and the fourth problem, denoted as Chemical, is about an optimization of simulated mass flow of Methyl Acetate ($d_\mathcal{X} = 1$ and $d_\Theta = 3$). Chemical has one constraint function for the upper-level problem. Energy and Chemical data are introduced by (Chew et al., 2025), in which these are regarded as a real-world dataset (see Chew et al. (2025) for the detailed definitions). From the fourth to the sixth problems, denoted as SMD01, 02, and 03 ($d_\mathcal{X} = 2$ and $d_\Theta = 2$), are test problems specifically designed for bilevel optimization benchmark (Sinha et al., 2014). The number of grid points in each dimension is 100 for GB and SB ($100^2$ points), and 10 for Energy and SMD ($10^4$ points).

The results are shown in Fig. 4. BLJES has obviously superior performance in BG, SB, Energy, and SMD02. For SMD01 and SMD03, similar performance is shown in BLJES and BILBO, both of which rapidly decrease the regret compared with Random.

### 5.3 DECOUPLED SETTING

We here evaluate performance on decoupled setting, for which regret comparison is shown in Fig. 5. The objective functions are the same functions used before. We see that MLJES shows smaller regret values for most of problems except only for SMD02 in which BILBO shows better performance. The results indicate that our MI based criterion is effective also for decoupled setting.

### 5.4 EFFECT OF THE NUMBER OF SAMPLINGS

We evaluate the effect of the number of samplings $K$ on the performance. Figure 6 shows the regret of BLJES with $K = 10, 20, 30,$ and 50 on the BG benchmark problem. Note that the result of $K = 30$ is same as Fig. 4 (a). Although $K = 50$ was slightly better than other settings in the end of the optimization, we do not see large differences. Similar tendency has been reported in information-theoretic BO studies (Wang & Jegelka, 2017; Takeno et al., 2022a). See in Appendix F.4 for the results on other problems.

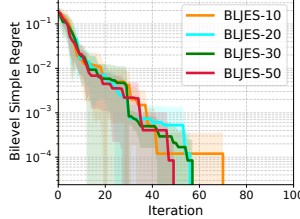

Figure 6: BLJES with different $K$ on the BG benchmark.

## 6 LIMITATIONS

As we described in section 1, the bilevel BO in which both levels are expensive has not been widely studied, and we still have several limitations. For example, it is widely known that BO has difficulty for dealing with high dimensional problems, though many studies have considered remedies to mitigate it (e.g., reviewer by Malu et al., 2021; González-Duque et al., 2024). Constructing those high dimensional specific strategies for bilevel BO is one of obviously important directions.

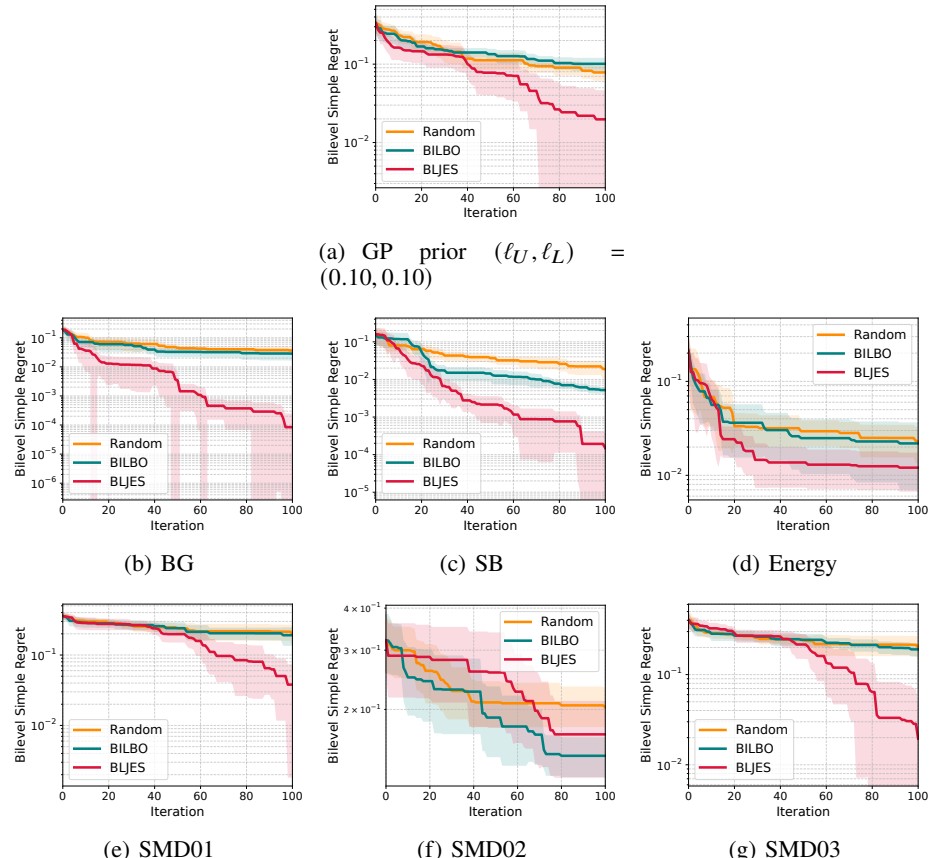

Figure 5: Regret comparison on benchmark problems under decoupled setting.

Another unresolved theme is the theoretical analysis of the approximation error of MI and the convergence of the optimization. The major approximation components of our MI estimator are the lower bound, RFF, and the MC sampling. Providing a theoretical guarantee for the combined approximation error is a challenging issue, though it is actually a common issue for information-theoretic BO methods. Particularly for GP-UCB based approaches including BILBO, the regret bound has been widely studied. On the other hand, for information-theoretic BO in general, the regret analysis is also still an open problem even for the simplest single-level standard problem setting (see Appendix G). Therefore, the regret analysis for information-theoretic BO including BLJES is still needed to be addressed.

## 7 CONCLUSION

We propose an information-theoretic approach to bilevel Bayesian optimization, called Bilevel optimization via Lower-bound based Joint Entropy Search (BLJES). BLJES considers information gain of optimal points and values of both the upper- and lower- level problems simultaneously, by which we can define a unified criterion that measures the benefit for both the problems. We derive a lower bound based approximation of bilevel information gain, which can be seen as a natural extension of the single level information-theoretic Bayesian optimization. Further, we also propose extensions for decoupled setting and constraint problems. The effectiveness of BLJES is demonstrated through sample path functions from Gaussian processes and benchmark functions.

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

# A    DERIVATION OF LOWER BOUND

## A.1    PROOF OF THEOREM 3.1

From Bayes theorem,

$$p(y^f_{(x,\theta)} \mid f_{(x,\theta^*(x))} \le f^*, \mathcal{D}_t^+) = \frac{p(f(x, \theta^*(x)) \le f^* \mid y^f_{(x,\theta)}, \mathcal{D}_t^+) p(y^f_{(x,\theta)} \mid \mathcal{D}_t^+)}{p(f(x, \theta^*(x)) \le f^* \mid \mathcal{D}_t^+)}. \tag{10}$$

All the three densities in the right hand side, the analytical representations can be derived as follows.

- The probability $p(f(x, \theta^*(x)) \le f^* \mid y^f_{(x,\theta)}, \mathcal{D}_t^+)$ is calculated by the density

$$f_{(x,\theta^*(x))} \mid y^f_{(x,\theta)}, \mathcal{D}_t^+ \sim \mathcal{N}(m_1^f, \{s_1^f\}^2),$$

for which the mean $m_1^f$ and variance $\{s_1^f\}^2$ can be derived by considering the conditional density of the joint posterior of $f_{(x,\theta^*(x))}, y^f_{(x,\theta)}$, and $f_{(x^*,\theta^*)}$ as

$$m_1^f = \mu_t^f(x, \theta^*(x))$$
$$+ \begin{bmatrix} \mathrm{Cov}_t^f((x, \theta^*(x)), (x, \theta)) \\ \mathrm{Cov}_t^f((x, \theta^*(x)), (x^*, \theta^*)) \end{bmatrix}^\top \begin{bmatrix} \{\sigma_t^f(x, \theta)\}^2 + \{\sigma_{\mathrm{noise}}^f\}^2 & \mathrm{Cov}_t^f((x, \theta), (x^*, \theta^*)) \\ \mathrm{Cov}_t^f((x^*, \theta^*), (x, \theta)) & \{\sigma_t^f(x^*, \theta^*)\}^2 \end{bmatrix}^{-1} \begin{bmatrix} y^f_{(x,\theta)} - \mu_t^f(x, \theta) \\ f_{(x^*,\theta^*)} - \mu_t^f(x^*, \theta^*) \end{bmatrix}$$

$$\{s_1^f\}^2 = \{\sigma_t^f(x, \theta^*(x))\}^2$$
$$- \begin{bmatrix} \mathrm{Cov}_t^f((x, \theta^*(x)), (x, \theta)) \\ \mathrm{Cov}_t^f((x, \theta^*(x)), (x^*, \theta^*)) \end{bmatrix}^\top \begin{bmatrix} \{\sigma_t^f(x, \theta)\}^2 + \{\sigma_{\mathrm{noise}}^f\}^2 & \mathrm{Cov}_t^f((x, \theta), (x^*, \theta^*)) \\ \mathrm{Cov}_t^f((x^*, \theta^*), (x, \theta)) & \{\sigma_t^f(x^*, \theta^*)\}^2 \end{bmatrix}^{-1} \begin{bmatrix} \mathrm{Cov}_t^f((x, \theta^*(x)), (x, \theta)) \\ \mathrm{Cov}_t^f((x, \theta^*(x)), (x^*, \theta^*)) \end{bmatrix}.$$

Note that $\mathrm{Cov}_t^f((x, \theta), (x', \theta'))$ is the posterior covariance between $f(x, \theta)$ and $f(x', \theta')$, given $\mathcal{D}_t$. By using $m_1^f$ and $s_1^f$, we have

$$p(f(x, \theta^*(x)) \le f^* \mid y^f_{(x,\theta)}, \mathcal{D}_t^+) = \begin{cases} \Phi\left(\frac{f^* - m_1^f}{s_1^f}\right) & \text{if } x \ne x^*, \\ 1 & \text{otherwise,} \end{cases} \tag{11}$$

where $\Phi$ is the cumulative density function (CDF) of the standard normal distribution.

- Next, to calculate the denominator $p(f_{(x,\theta^*(x))} \le f^* \mid \mathcal{D}_t^+)$, we consider the density

$$f_{(x,\theta^*(x))} \mid \mathcal{D}_t^+ \sim \mathcal{N}(m_2^f, \{s_2^f\}^2),$$

for which the mean $m_2^f$ and variance $\{s_2^f\}^2$ can be derived by considering the conditional density of the joint posterior of $f_{(x, \theta^*(x))}$ and $f_{(x^*, \theta^*)}$ as

$$m_2^f = \mu_t^f(x, \theta^*(x)) + \frac{\text{Cov}_t^f((x, \theta^*(x)), (x^*, \theta^*))}{\left\{\sigma_t^f(x^*, \theta^*)\right\}^2}(f(x^*, \theta^*) - \mu_t^f(x^*, \theta^*))$$

$$\{s_2^f\}^2 = \sigma_t^f(x, \theta^*(x)) - \frac{\left\{\text{Cov}_t^f((x, \theta^*(x)), (x^*, \theta^*))\right\}^2}{\left\{\sigma_t^f(x^*, \theta^*)\right\}^2}$$

Then, we obtain

$$p(f_{(x, \theta^*(x))} \le f^* \mid \mathcal{D}_t^+) = \begin{cases} \Phi\left(\frac{f^* - m_2^f}{s_2^f}\right) & \text{if } x \ne x^* \\ 1 & \text{otherwise .} \end{cases} \tag{12}$$

- The density $p(y_{(x, \theta)}^f \mid \mathcal{D}_t^+)$ can also be derived by a similar approach as

$$y_{(x, \theta)}^f \mid \mathcal{D}_t^+ \sim \mathcal{N}(m_3^f, s_3^f),$$

where

$$m_3^f = \mu_t^f(x, \theta) + \frac{\text{Cov}_t^f((x, \theta), (x^*, \theta^*))}{\left\{\sigma_t^f(x^*, \theta^*)\right\}^2}(f(x^*, \theta^*) - \mu_t^f(x^*, \theta^*))$$

$$\{s_3^f\}^2 = \{\sigma_t^f(x, \theta)\}^2 + \{\sigma_{\text{noise}}^f\}^2 - \frac{\left\{\text{Cov}_t^f((x, \theta), (x^*, \theta^*))\right\}^2}{\left\{\sigma_t^f(x^*, \theta^*)\right\}^2}.$$

Therefore,

$$p(y_{(x, \theta)}^f \mid \mathcal{D}_t^+) = \phi\left(\frac{y_{(x, \theta)}^f - m_3^f}{s_3^f}\right)/s_3^f. \tag{13}$$

where $\phi$ is the density function of the standard normal distribution.

By substituting (11), (12), and (13) into (10), we obtain (6).

## A.2 Analytical Representation of $p(y_{(x, \theta)}^g \mid g(x^*, \theta) \le g^*, \mathcal{D}_t^+)$

In the case of $p(y_{(x, \theta)}^g \mid g(x^*, \theta) \le g^*, \mathcal{D}_t^+)$, almost the same derivation can be applied as (6). Therefore, we here only show the final result

$$p(y_{(x, \theta)}^g \mid g(x^*, \theta) \le g^*, \mathcal{D}_t^+) = \begin{cases} \Phi\left(\frac{g^* - m_1^g}{s_1^g}\right) \phi\left(\frac{y_{(x, \theta)}^g - m_3^g}{s_3^g}\right)/\Phi\left(\frac{g^* - m_2^g}{s_2^g}\right) s_3^g & \text{if } \theta \ne \theta^*, \\ \phi\left(\frac{y_{(x, \theta)}^g - m_3^g}{s_3^g}\right)/s_3^g & \text{otherwise,} \end{cases}$$

where

$$m_1^g = \mu_t^g(\boldsymbol{x}^*, \boldsymbol{\theta}) + \begin{bmatrix} \mathrm{Cov}_t^g((\boldsymbol{x}^*, \boldsymbol{\theta}), (\boldsymbol{x}, \boldsymbol{\theta})) \\ \mathrm{Cov}_t^g((\boldsymbol{x}^*, \boldsymbol{\theta}), (\boldsymbol{x}^*, \boldsymbol{\theta}^*)) \end{bmatrix}^\top \begin{bmatrix} \{\sigma_t^g(\boldsymbol{x}, \boldsymbol{\theta})\}^2 + \{\sigma_{\mathrm{noise}}^g\}^2 & \mathrm{Cov}_t^g((\boldsymbol{x}, \boldsymbol{\theta}), (\boldsymbol{x}^*, \boldsymbol{\theta}^*)) \\ \mathrm{Cov}_t^g((\boldsymbol{x}^*, \boldsymbol{\theta}^*), (\boldsymbol{x}, \boldsymbol{\theta})) & \{\sigma_t^g(\boldsymbol{x}^*, \boldsymbol{\theta}^*)\}^2 \end{bmatrix}^{-1} \begin{bmatrix} y_{(\boldsymbol{x}, \boldsymbol{\theta})}^g - \mu_t^g(\boldsymbol{x}, \boldsymbol{\theta}) \\ g_{(\boldsymbol{x}^*, \boldsymbol{\theta}^*)} - \mu_t^g(\boldsymbol{x}^*, \boldsymbol{\theta}^*) \end{bmatrix},$$

$$\{s_1^g\}^2 = \{\sigma_t^g(\boldsymbol{x}^*, \boldsymbol{\theta})\}^2$$
$$\qquad - \begin{bmatrix} \mathrm{Cov}_t^g((\boldsymbol{x}^*, \boldsymbol{\theta}), (\boldsymbol{x}, \boldsymbol{\theta})) \\ \mathrm{Cov}_t^g((\boldsymbol{x}^*, \boldsymbol{\theta}), (\boldsymbol{x}^*, \boldsymbol{\theta}^*)) \end{bmatrix}^\top \begin{bmatrix} \{\sigma_t^g(\boldsymbol{x}, \boldsymbol{\theta})\}^2 + \{\sigma_{\mathrm{noise}}^g\}^2 & \mathrm{Cov}_t^f((\boldsymbol{x}, \boldsymbol{\theta}), (\boldsymbol{x}^*, \boldsymbol{\theta}^*)) \\ \mathrm{Cov}_t^g((\boldsymbol{x}^*, \boldsymbol{\theta}^*), (\boldsymbol{x}, \boldsymbol{\theta})) & \{\sigma_t^g(\boldsymbol{x}^*, \boldsymbol{\theta}^*)\}^2 \end{bmatrix}^{-1} \begin{bmatrix} \mathrm{Cov}_t^g((\boldsymbol{x}^*, \boldsymbol{\theta}), (\boldsymbol{x}, \boldsymbol{\theta})) \\ \mathrm{Cov}_t^g((\boldsymbol{x}^*, \boldsymbol{\theta}), (\boldsymbol{x}^*, \boldsymbol{\theta}^*)) \end{bmatrix},$$

$$m_2^g = \mu_t^g(\boldsymbol{x}^*, \boldsymbol{\theta}) + \frac{\mathrm{Cov}_t^g((\boldsymbol{x}^*, \boldsymbol{\theta}), (\boldsymbol{x}^*, \boldsymbol{\theta}^*))}{\left\{\sigma_t^g(\boldsymbol{x}^*, \boldsymbol{\theta}^*)\right\}^2}(g(\boldsymbol{x}^*, \boldsymbol{\theta}^*) - \mu_t^g(\boldsymbol{x}^*, \boldsymbol{\theta}^*))$$

$$\{s_2^g\}^2 = \{\sigma_t^g(\boldsymbol{x}^*, \boldsymbol{\theta})\}^2 - \frac{\left\{\mathrm{Cov}_t^g((\boldsymbol{x}^*, \boldsymbol{\theta}), (\boldsymbol{x}^*, \boldsymbol{\theta}^*))\right\}^2}{\left\{\sigma_t^g(\boldsymbol{x}^*, \boldsymbol{\theta}^*)\right\}^2},$$

$$m_3^g = \mu_t^g(\boldsymbol{x}, \boldsymbol{\theta}) + \frac{\mathrm{Cov}_t^g((\boldsymbol{x}, \boldsymbol{\theta}), (\boldsymbol{x}^*, \boldsymbol{\theta}^*))}{\left\{\sigma_t^g(\boldsymbol{x}^*, \boldsymbol{\theta}^*)\right\}^2}(g(\boldsymbol{x}^*, \boldsymbol{\theta}^*) - \mu_t^g(\boldsymbol{x}^*, \boldsymbol{\theta}^*)),$$

$$\{s_3^g\}^2 = \{\sigma_t^g(\boldsymbol{x}, \boldsymbol{\theta})\}^2 + \{\sigma_{\mathrm{noise}}^g\}^2 - \frac{\left\{\mathrm{Cov}_t^g((\boldsymbol{x}, \boldsymbol{\theta}), (\boldsymbol{x}^*, \boldsymbol{\theta}^*))\right\}^2}{\left\{\sigma_t^g(\boldsymbol{x}^*, \boldsymbol{\theta}^*)\right\}^2},$$

and $\mathrm{Cov}_t^g((\boldsymbol{x}, \boldsymbol{\theta}), (\boldsymbol{x}', \boldsymbol{\theta}'))$ is the posterior covariance between $g(\boldsymbol{x}, \boldsymbol{\theta})$ and $g(\boldsymbol{x}', \boldsymbol{\theta}')$ given $\mathcal{D}_t$.

# B  LOWER BOUND AS INDEPENDENCE APPROXIMATION

## B.1  DERIVING BLJES THROUGH INDEPENDENCE APPROXIMATION

Here, we show that our variational distribution (4) can be interpreted through a conditional independence approximation of the GPs. In other words, under the assumption of the conditional independence, the lower bound (3) becomes equal to the original MI. In a variational inference, it is common to introduce independence assumptions for making computations tractable. Our truncation based computations can also be interpreted by a similar perspective that has not been revealed in other studies of information-theoretic BO.

The variational distribution $q(y_{(\boldsymbol{x}, \boldsymbol{\theta})}^f, y_{(\boldsymbol{x}, \boldsymbol{\theta})}^g \mid f^*, g^*, \boldsymbol{x}^*, \boldsymbol{\theta}^*, \mathcal{D}_t)$ is a surrogate for $p(y_{(\boldsymbol{x}, \boldsymbol{\theta})}^f, y_{(\boldsymbol{x}, \boldsymbol{\theta})}^g \mid f^*, g^*, \boldsymbol{x}^*, \boldsymbol{\theta}^*, \mathcal{D}_t)$. The conditioning of $f^*, g^*, \boldsymbol{x}^*$ and $\boldsymbol{\theta}^*$ can be represented by the following three conditions:

C1. When $f^*$ is given, $f(\boldsymbol{x}', \boldsymbol{\theta}^*(\boldsymbol{x}')) \le f^*$ should hold for $\forall \boldsymbol{x}'$.

C2. When $g^*$ is given, $g(\boldsymbol{x}^*, \boldsymbol{\theta}') \le g^*$ should hold for $\forall \boldsymbol{\theta}'$.

C3. $f(\boldsymbol{x}^*, \boldsymbol{\theta}^*) = f^*$ and $g(\boldsymbol{x}^*, \boldsymbol{\theta}^*) = g^*$

The condition C3 can be realized just by adding $(\boldsymbol{x}^*, \boldsymbol{\theta}^*, f^*, g^*)$ into the training dataset $\mathcal{D}_t$, i.e., conditioning on $\mathcal{D}_t^+$.

The posterior distribution over the 'entire functions' $f$ and $g$ (not only for given specific input points) with all the above three conditions is represented as

$$p(f, g \mid f^*, g^*, \boldsymbol{x}^*, \boldsymbol{\theta}^*, \mathcal{D}_t) = p(f \mid \mathcal{D}_t^+) p(g \mid \mathcal{D}_t^+) \times$$
$$\mathbb{I}(f(\boldsymbol{x}', \boldsymbol{\theta}^*(\boldsymbol{x}')) \le f^* \text{ for } \forall \boldsymbol{x}') \, \mathbb{I}(g(\boldsymbol{x}^*, \boldsymbol{\theta}') \le g^* \text{ for } \forall \boldsymbol{\theta}') / Z, \tag{14}$$

where $\mathbb{I}$ is the indicator function and $Z$ is a normalizing constant. In this density, the term $p(f \mid \mathcal{D}_t^+) p(g \mid \mathcal{D}_t^+)$ generates functions satisfying the condition C3. The two indicator functions only accept functions satisfying the conditions C1 and C2. The predictive distribution $p(f(\boldsymbol{x}, \boldsymbol{\theta}), g(\boldsymbol{x}, \boldsymbol{\theta}) \mid f^*, g^*, \boldsymbol{x}^*, \boldsymbol{\theta}^*, \mathcal{D}_t)$ is obtained by marginalizing the above distribution over all $f(\boldsymbol{x}', \boldsymbol{\theta}')$ and $g(\boldsymbol{x}', \boldsymbol{\theta}')$ such that $(\boldsymbol{x}', \boldsymbol{\theta}') \ne (\boldsymbol{x}, \boldsymbol{\theta})$. However, this marginalization is obviously computationally intractable.

To simplify the marginalization, we introduce the following conditional independence approximation

$$p(f \mid \mathcal{D}_t^+) \approx p(f(\boldsymbol{x}, \boldsymbol{\theta}), f(\boldsymbol{x}, \boldsymbol{\theta}^*(\boldsymbol{x})) \mid \mathcal{D}_t^+) \, p(f' \mid \mathcal{D}_t^+), \tag{15}$$

$$p(g \mid \mathcal{D}_t^+) \approx p(g(\boldsymbol{x}, \boldsymbol{\theta}), g(\boldsymbol{x}^*, \boldsymbol{\theta}) \mid \mathcal{D}_t^+) \, p(g' \mid \mathcal{D}_t^+), \tag{16}$$

where $f'$ is $f$ in which $(\boldsymbol{x}, \boldsymbol{\theta})$ and $(\boldsymbol{x}, \boldsymbol{\theta}^*(\boldsymbol{x}))$ are removed from its input domain, and $g'$ is $g$ in which $(\boldsymbol{x}, \boldsymbol{\theta})$ and $(\boldsymbol{x}^*, \boldsymbol{\theta})$ are removed from its input domain. This approximation means that, for the joint GP posteriors given $\mathcal{D}_t^+$, the posterior covariances between $f'$ and $\{f(\boldsymbol{x}, \boldsymbol{\theta}), f(\boldsymbol{x}, \boldsymbol{\theta}^*(\boldsymbol{x}))\}$, and between $g'$ and $\{g(\boldsymbol{x}, \boldsymbol{\theta}), g(\boldsymbol{x}^*, \boldsymbol{\theta})\}$ are regarded as 0. By substituting (15) and (16) into (14), we obtain

$$
\begin{aligned}
& p(f(\boldsymbol{x}, \boldsymbol{\theta}), f(\boldsymbol{x}, \boldsymbol{\theta}^*(\boldsymbol{x})) \mid \mathcal{D}_t^+) p(f' \mid \mathcal{D}_t^+) \, p(g(\boldsymbol{x}, \boldsymbol{\theta}), g(\boldsymbol{x}^*, \boldsymbol{\theta}) \mid \mathcal{D}_t^+) p(g' \mid \mathcal{D}_t^+) \times \\
& \mathbb{I}(f(\boldsymbol{x}', \boldsymbol{\theta}^*(\boldsymbol{x}')) \le f^* \text{ for } \forall \boldsymbol{x}') \, \mathbb{I}(g(\boldsymbol{x}^*, \boldsymbol{\theta}') \le g^* \text{ for } \forall \boldsymbol{\theta}') \, / \, Z.
\end{aligned} \tag{17}
$$

To derive the predictive distribution $p(f(\boldsymbol{x}, \boldsymbol{\theta}), g(\boldsymbol{x}, \boldsymbol{\theta}) \mid f^*, g^*, \boldsymbol{x}^*, \boldsymbol{\theta}^*, \mathcal{D}_t)$ under the independence assumption, we consider the marginalization of the approximate distribution (17) over $\mathcal{X} \times \Theta$ except for $(\boldsymbol{x}, \boldsymbol{\theta})$. For $f$, this marginalization can be seen as the joint marginalization of $f(\boldsymbol{x}, \boldsymbol{\theta}^*(\boldsymbol{x}))$ and $f'$, and for $g$, can be seen as the joint marginalization of $g(\boldsymbol{x}^*, \boldsymbol{\theta})$ and $g'$. By combining (17) and the decomposition of the indicator functions

$$
\mathbb{I}(f(\boldsymbol{x}', \boldsymbol{\theta}^*(\boldsymbol{x}')) \le f^* \text{ for } \forall \boldsymbol{x}') = \mathbb{I}(f(\boldsymbol{x}, \boldsymbol{\theta}^*(\boldsymbol{x})) \le f^*) \, \mathbb{I}(f'(\boldsymbol{x}', \boldsymbol{\theta}^*(\boldsymbol{x}')) \le f^* \text{ for } \forall \boldsymbol{x}' \ne \boldsymbol{x}),
$$

$$
\mathbb{I}(g(\boldsymbol{x}^*, \boldsymbol{\theta}') \le g^* \text{ for } \forall \boldsymbol{\theta}') = \mathbb{I}(g(\boldsymbol{x}^*, \boldsymbol{\theta}) \le g^*) \, \mathbb{I}(g'(\boldsymbol{x}^*, \boldsymbol{\theta}') \le g^* \text{ for } \forall \boldsymbol{\theta}' \ne \boldsymbol{\theta}),
$$

we obtain

$$
p(f(\boldsymbol{x}, \boldsymbol{\theta}), g(\boldsymbol{x}, \boldsymbol{\theta}) \mid f^*, g^*, \boldsymbol{x}^*, \boldsymbol{\theta}^*, \mathcal{D}_t)
$$

$$
= \int_{f(\boldsymbol{x}, \boldsymbol{\theta}^*(\boldsymbol{x}))} \int_{g(\boldsymbol{x}^*, \boldsymbol{\theta})} \int_{f'} \int_{g'} p(f, g \mid f^*, g^*, \boldsymbol{x}^*, \boldsymbol{\theta}^*, \mathcal{D}_t) \, \mathrm{d}g' \mathrm{d}f' \mathrm{d}g(\boldsymbol{x}^*, \boldsymbol{\theta}) \mathrm{d}f(\boldsymbol{x}, \boldsymbol{\theta}^*(\boldsymbol{x}))
$$

$$
\approx \underbrace{\int_{f(\boldsymbol{x}, \boldsymbol{\theta}^*(\boldsymbol{x}))} p(f(\boldsymbol{x}, \boldsymbol{\theta}), f(\boldsymbol{x}, \boldsymbol{\theta}^*(\boldsymbol{x})) \mid \mathcal{D}_t^+) \mathbb{I}(f(\boldsymbol{x}, \boldsymbol{\theta}^*(\boldsymbol{x})) \le f^*) \mathrm{d}f(\boldsymbol{x}, \boldsymbol{\theta}^*(\boldsymbol{x})) \times}_{A}
$$

$$
\underbrace{\int_{g(\boldsymbol{x}^*, \boldsymbol{\theta})} p(g(\boldsymbol{x}, \boldsymbol{\theta}), g(\boldsymbol{x}^*, \boldsymbol{\theta}) \mid \mathcal{D}_t^+) \mathbb{I}(g(\boldsymbol{x}^*, \boldsymbol{\theta}) \le g^*) \mathrm{d}g(\boldsymbol{x}^*, \boldsymbol{\theta}) \times}_{B}
$$

$$
\underbrace{\int_{f'} \int_{g'} p(f' \mid \mathcal{D}_t^+) \mathbb{I}(f'(\boldsymbol{x}', \boldsymbol{\theta}^*(\boldsymbol{x}')) \le f^* \text{ for } \forall \boldsymbol{x}' \ne \boldsymbol{x}) \, p(g' \mid \mathcal{D}_t^+) \mathbb{I}(g'(\boldsymbol{x}^*, \boldsymbol{\theta}') \le g^* \text{ for } \forall \boldsymbol{\theta}' \ne \boldsymbol{\theta}) \, / \, Z \, \mathrm{d}g' \mathrm{d}f'}_{C}
$$

$$
= p(f(\boldsymbol{x}, \boldsymbol{\theta}) \mid f(\boldsymbol{x}, \boldsymbol{\theta}^*(\boldsymbol{x})) \le f^*, \mathcal{D}_t^+) \, p(g(\boldsymbol{x}, \boldsymbol{\theta}) \mid g(\boldsymbol{x}^*, \boldsymbol{\theta}) \le g^*, \mathcal{D}_t^+), \tag{18}
$$

where $\int_{f'}$, $\mathrm{d}f'$ and $\int_{g'}$, $\mathrm{d}g'$ are marginalization over the entire input domain of $f'$ and $g'$. The term $C$ is the constant factor independent from $f(\boldsymbol{x}, \boldsymbol{\theta})$ and $g(\boldsymbol{x}, \boldsymbol{\theta})$. Then, the last line is derived from the normalizing condition of the density, and the relations $A \propto p(f(\boldsymbol{x}, \boldsymbol{\theta}) \mid f(\boldsymbol{x}, \boldsymbol{\theta}^*(\boldsymbol{x})) \le f^*, \mathcal{D}_t^+)$ and $B \propto p(g(\boldsymbol{x}, \boldsymbol{\theta}) \mid g(\tilde{\boldsymbol{x}}^*, \boldsymbol{\theta}) \le g^*, \mathcal{D}_t^+)$.

Then, by using (18), $p(y_{(\boldsymbol{x}, \boldsymbol{\theta})}^f, y_{(\boldsymbol{x}, \boldsymbol{\theta})}^g \mid f^*, g^*, \boldsymbol{x}^*, \boldsymbol{\theta}^*, \mathcal{D}_t)$ can be represented as

$$
p(y_{(\boldsymbol{x}, \boldsymbol{\theta})}^f, y_{(\boldsymbol{x}, \boldsymbol{\theta})}^g \mid f^*, g^*, \boldsymbol{x}^*, \boldsymbol{\theta}^*, \mathcal{D}_t)
$$

$$
= p(y_{(\boldsymbol{x}, \boldsymbol{\theta})}^f \mid f(\boldsymbol{x}, \boldsymbol{\theta})) p(y_{(\boldsymbol{x}, \boldsymbol{\theta})}^g \mid g(\boldsymbol{x}, \boldsymbol{\theta})) \, p(f(\boldsymbol{x}, \boldsymbol{\theta}), g(\boldsymbol{x}, \boldsymbol{\theta}) \mid f^*, g^*, \boldsymbol{x}^*, \boldsymbol{\theta}^*, \mathcal{D}_t)
$$

$$
\approx p(y_{(\boldsymbol{x}, \boldsymbol{\theta})}^f \mid f(\boldsymbol{x}, \boldsymbol{\theta})) p(y_{(\boldsymbol{x}, \boldsymbol{\theta})}^g \mid g(\boldsymbol{x}, \boldsymbol{\theta})) p(f(\boldsymbol{x}, \boldsymbol{\theta}) \mid f(\boldsymbol{x}, \boldsymbol{\theta}^*(\boldsymbol{x})) \le f^*, \mathcal{D}_t^+) \, p(g(\boldsymbol{x}, \boldsymbol{\theta}) \mid g(\boldsymbol{x}^*, \boldsymbol{\theta}) \le g^*, \mathcal{D}_t^+)
$$

$$
= p(y_{(\boldsymbol{x}, \boldsymbol{\theta})}^f \mid f(\boldsymbol{x}, \boldsymbol{\theta}^*(\boldsymbol{x})) \le f^*, \mathcal{D}_t^+) \, p(y_{(\boldsymbol{x}, \boldsymbol{\theta})}^g \mid g(\boldsymbol{x}^*, \boldsymbol{\theta}) \le g^*, \mathcal{D}_t^+)
$$

$$
= q(y_{(\boldsymbol{x}, \boldsymbol{\theta})}^f, y_{(\boldsymbol{x}, \boldsymbol{\theta})}^g \mid f^*, g^*, \boldsymbol{x}^*, \boldsymbol{\theta}^*, \mathcal{D}_t).
$$

This indicates that if the independence assumption (15) and (16) hold (if the posterior is actually independent), the variational distribution becomes the true distribution, resulting in that the lower bound (3) becomes equal to the original MI. On the other hand, clarifying the error caused by this independence assumption for the general dependent case is still future work. However, this analysis revealed that our variational distribution (3) can be purely derived from the independence assumption without manually specifying a particular form of a distribution. To our knowledge, no information-theoretic BO studies have revealed this way of justification for the truncate distribution approximation.

## B.2 Deriving Simplified BLJES without Truncation

One of notable properties of BLJES is to consider the two truncations $f(\boldsymbol{x}, \boldsymbol{\theta}^*(\boldsymbol{x})) \leq f^*$ and $g(\boldsymbol{x}^*, \boldsymbol{\theta}) \leq g^*$ as described in the first and the second items in the itemization after (4). Intuitively, these conditions transmit the information that $f^*$ and $g^*$ are the maximum values of the upper- and the lower- problems to the query point $(\boldsymbol{x}, \boldsymbol{\theta})$. On the other hand, by introducing stronger independence assumptions than (15) and (16), we can derive a simplified variant of BLJES, which we use for empirically evaluating the importance of truncations in Appendix F.8.

In this simplified variant, instead of (15) and (16), the conditional independence is assumed between $(\boldsymbol{x}, \boldsymbol{\theta})$ and all the other input points, which results in

$$p(f \mid \mathcal{D}_t^+) \approx p(f(\boldsymbol{x}, \boldsymbol{\theta}) \mid \mathcal{D}_t^+) \, p(f' \mid \mathcal{D}_t^+), \tag{19}$$

$$p(g \mid \mathcal{D}_t^+) \approx p(g(\boldsymbol{x}, \boldsymbol{\theta}) \mid \mathcal{D}_t^+) \, p(g' \mid \mathcal{D}_t^+), \tag{20}$$

where $f'$ and $g'$ are $f$ and $g$ in which only $(\boldsymbol{x}, \boldsymbol{\theta})$ is removed from the input domain (note that the definitions are slightly different from the case of (15) and (16)). By the almost same derivation in the previous Appendix B.1, we obtain

$$p(y_{(\boldsymbol{x},\boldsymbol{\theta})}^f, y_{(\boldsymbol{x},\boldsymbol{\theta})}^g \mid f^*, g^*, \boldsymbol{x}^*, \boldsymbol{\theta}^*, \mathcal{D}_t) \approx p(y_{(\boldsymbol{x},\boldsymbol{\theta})}^f \mid \mathcal{D}_t^+) \, p(y_{(\boldsymbol{x},\boldsymbol{\theta})}^g \mid \mathcal{D}_t^+).$$

We can define the acquisition function by defining $q(y_{(\boldsymbol{x},\boldsymbol{\theta})}^f, y_{(\boldsymbol{x},\boldsymbol{\theta})}^g \mid f^*, g^*, \boldsymbol{x}^*, \boldsymbol{\theta}^*, \mathcal{D}_t)$ as the right hand side of this approximation. As a result, we obtain a simper lower bound (a lower bound without truncation)

$$\mathrm{LB}_{\mathrm{wot}}(\boldsymbol{x}, \boldsymbol{\theta}) := \mathbb{E}_{\Omega}\left[\log \frac{p(y_{(\boldsymbol{x},\boldsymbol{\theta})}^f \mid \mathcal{D}_t^+)}{p(y_{(\boldsymbol{x},\boldsymbol{\theta})}^f \mid \mathcal{D}_t)} + \log \frac{p(y_{(\boldsymbol{x},\boldsymbol{\theta})}^g \mid \mathcal{D}_t^+)}{p(y_{(\boldsymbol{x},\boldsymbol{\theta})}^g \mid \mathcal{D}_t)}\right], \tag{21}$$

for which the same Monte-Carlo approximation as the original BLJES can be applied. By comparing $\mathrm{LB}_{\mathrm{wot}}(\boldsymbol{x}, \boldsymbol{\theta})$ with (5), we see that the truncation conditions are removed.

## C Detail of Gradient Computations

First, we consider the gradient for $\partial \tilde{f}(\boldsymbol{x}, \boldsymbol{\theta}^*(\boldsymbol{x}))/\partial \boldsymbol{x}$, which is required to obtain the sample of $\boldsymbol{x}^*, \boldsymbol{\theta}^*, f^*$, and $g^*$. For $\tilde{\boldsymbol{\theta}}^*(\boldsymbol{x}) = \arg\max_{\boldsymbol{\theta}} \tilde{g}(\boldsymbol{x}, \boldsymbol{\theta})$, the implicit function theorem derives

$$\frac{\partial \tilde{\boldsymbol{\theta}}^*(\boldsymbol{x})}{\partial \boldsymbol{x}^\top} = -\left\{ \left.\frac{\partial^2 \tilde{g}(\boldsymbol{x}, \boldsymbol{\theta})}{\partial \boldsymbol{\theta} \partial \boldsymbol{\theta}^\top}\right|_{\boldsymbol{\theta} = \boldsymbol{\theta}^*(\boldsymbol{x})} \right\}^{-1} \left.\frac{\partial^2 \tilde{g}(\boldsymbol{x}, \boldsymbol{\theta})}{\partial \boldsymbol{\theta} \partial \boldsymbol{x}^\top}\right|_{\boldsymbol{\theta} = \boldsymbol{\theta}^*(\boldsymbol{x})},$$

from which we can calculate

$$\frac{\partial \tilde{f}(\boldsymbol{x}, \tilde{\boldsymbol{\theta}}^*(\boldsymbol{x}))}{\partial \boldsymbol{x}} = \left.\frac{\partial \tilde{f}(\boldsymbol{x}, \boldsymbol{\theta})}{\partial \boldsymbol{x}}\right|_{\boldsymbol{\theta} = \tilde{\boldsymbol{\theta}}^*(\boldsymbol{x})} + \left\{\frac{\partial \tilde{\boldsymbol{\theta}}^*(\boldsymbol{x})}{\partial \boldsymbol{x}^\top}\right\}^\top \left.\frac{\partial \tilde{f}(\boldsymbol{x}, \boldsymbol{\theta})}{\partial \boldsymbol{\theta}}\right|_{\boldsymbol{\theta} = \tilde{\boldsymbol{\theta}}^*(\boldsymbol{x})}.$$

Next, we consider the acquisition function maximization. Let

$$\tilde{a}(\boldsymbol{x}, \boldsymbol{\theta}, \boldsymbol{\theta}') := \log \frac{p(\tilde{y}_{(\boldsymbol{x},\boldsymbol{\theta})}^f \mid \tilde{f}(\boldsymbol{x}, \boldsymbol{\theta}') \leq \tilde{f}^*, \tilde{\mathcal{D}}_t^+)}{p(\tilde{y}_{(\boldsymbol{x},\boldsymbol{\theta})}^f \mid \mathcal{D}_t)} + \log \frac{p(\tilde{y}_{(\boldsymbol{x},\boldsymbol{\theta})}^g \mid \tilde{g}(\tilde{\boldsymbol{x}}^*, \boldsymbol{\theta}) \leq \tilde{g}^*, \tilde{\mathcal{D}}_t^+)}{p(\tilde{y}_{(\boldsymbol{x},\boldsymbol{\theta})}^g \mid \mathcal{D}_t)}$$

be the inside of the expectation of (5) in which $\boldsymbol{\theta}^*(\boldsymbol{x})$ is replaced with $\boldsymbol{\theta}'$, and variables in $\Omega$ is replaced by a sample, denoted with "~". Note that $\tilde{D}_t^+ = \mathcal{D}_t \cup (\tilde{\boldsymbol{x}}^*, \tilde{\boldsymbol{\theta}}^*, \tilde{f}^*, \tilde{g}^*)$. Then, the gradient with respect to $\boldsymbol{x}$ can be written as

$$\frac{\partial \tilde{a}(\boldsymbol{x}, \boldsymbol{\theta}, \tilde{\boldsymbol{\theta}}^*(\boldsymbol{x}))}{\partial \boldsymbol{x}} = \left.\frac{\partial \tilde{a}(\boldsymbol{x}, \boldsymbol{\theta}, \boldsymbol{\theta}')}{\partial \boldsymbol{x}}\right|_{\boldsymbol{\theta}' = \tilde{\boldsymbol{\theta}}^*(\boldsymbol{x})} + \left\{\frac{\partial \tilde{\boldsymbol{\theta}}^*(\boldsymbol{x})}{\partial \boldsymbol{x}^\top}\right\}^\top \left.\frac{\partial \tilde{a}(\boldsymbol{x}, \boldsymbol{\theta}, \boldsymbol{\theta}')}{\partial \boldsymbol{\theta}'}\right|_{\boldsymbol{\theta}' = \tilde{\boldsymbol{\theta}}^*(\boldsymbol{x})}$$

The gradient with respect $\boldsymbol{\theta}$ can be obtained through the usual derivative.

## D  LOWER BOUND OF DECOUPLED SETTING

The lower bound of the information gain for the upper-level observation is

$$\text{MI}(y^f_{(\boldsymbol{x},\boldsymbol{\theta})} \, ; \, f^*, g^*, \boldsymbol{x}^*, \boldsymbol{\theta}^* \mid \mathcal{D}_t) = \mathbb{E}_{\Omega}\left[\log \frac{p(y^f_{(\boldsymbol{x},\boldsymbol{\theta})} \mid f^*, g^*, \boldsymbol{x}^*, \boldsymbol{\theta}^*, \mathcal{D}_t)}{p(y^f_{(\boldsymbol{x},\boldsymbol{\theta})} \mid \mathcal{D}_t)}\right]$$

$$= \mathbb{E}_{f^*, g^*, \boldsymbol{x}^*, \boldsymbol{\theta}^*}\left[\mathbb{E}_{y^f_{(\boldsymbol{x},\boldsymbol{\theta})} \mid f^*, g^*, \boldsymbol{x}^*, \boldsymbol{\theta}^*, \mathcal{D}_t}\left[\log \frac{q(y^f_{(\boldsymbol{x},\boldsymbol{\theta})} \mid f^*, g^*, \boldsymbol{x}^*, \boldsymbol{\theta}^*, \mathcal{D}_t)}{p(y^f_{(\boldsymbol{x},\boldsymbol{\theta})} \mid \mathcal{D}_t)}\right]\right.$$

$$\left. + \text{KL}\left(p(y^f_{(\boldsymbol{x},\boldsymbol{\theta})} \mid f^*, g^*, \boldsymbol{x}^*, \boldsymbol{\theta}^*, \mathcal{D}_t) \parallel q(y^f_{(\boldsymbol{x},\boldsymbol{\theta})} \mid f^*, g^*, \boldsymbol{x}^*, \boldsymbol{\theta}^*, \mathcal{D}_t)\right)\right]$$

$$\geq \mathbb{E}_{\Omega}\left[\log \frac{q(y^f_{(\boldsymbol{x},\boldsymbol{\theta})} \mid f^*, g^*, \boldsymbol{x}^*, \boldsymbol{\theta}^*, \mathcal{D}_t)}{p(y^f_{(\boldsymbol{x},\boldsymbol{\theta})} \mid \mathcal{D}_t)}\right] =: \text{LB}^f(\boldsymbol{x}, \boldsymbol{\theta}).$$

By setting the variational distribution as

$$q(y^f_{(\boldsymbol{x},\boldsymbol{\theta})} \mid f^*, g^*, \boldsymbol{x}^*, \boldsymbol{\theta}^*, \mathcal{D}_t) := p(y^f_{(\boldsymbol{x},\boldsymbol{\theta})} \mid f(\boldsymbol{x}, \boldsymbol{\theta}^*(\boldsymbol{x})) \leq f^*, g(\boldsymbol{x}^*, \boldsymbol{\theta}) \leq g^*, \mathcal{D}_t^+)$$

$$= p(y^f_{(\boldsymbol{x},\boldsymbol{\theta})} \mid f(\boldsymbol{x}, \boldsymbol{\theta}^*(\boldsymbol{x})) \leq f^*, \mathcal{D}_t^+),$$

we obtain the lower bound (7).

## E  EXTENSION FOR CONSTRAINT PROBLEMS

Let

$$\boldsymbol{h}^f_{(\boldsymbol{x},\boldsymbol{\theta})} := (f(\boldsymbol{x}, \boldsymbol{\theta}), c^U_1(\boldsymbol{x}, \boldsymbol{\theta}), \ldots, c^U_N(\boldsymbol{x}, \boldsymbol{\theta}))^\top,$$

$$\boldsymbol{h}^g_{(\boldsymbol{x},\boldsymbol{\theta})} := (g(\boldsymbol{x}, \boldsymbol{\theta}), c^L_1(\boldsymbol{x}, \boldsymbol{\theta}), \ldots, c^L_M(\boldsymbol{x}, \boldsymbol{\theta}))^\top,$$

be the vectors in which the objective function and the constraint functions are concatenated for the upper- and the lower-level problems, respectively, and

$$\boldsymbol{y}^f_{(\boldsymbol{x},\boldsymbol{\theta})} := (y^f_{(\boldsymbol{x},\boldsymbol{\theta})}, y^{c^U_1}_{(\boldsymbol{x},\boldsymbol{\theta})}, \ldots, y^{c^U_N}_{(\boldsymbol{x},\boldsymbol{\theta})})^\top,$$

$$\boldsymbol{y}^g_{(\boldsymbol{x},\boldsymbol{\theta})} := (y^g_{(\boldsymbol{x},\boldsymbol{\theta})}, y^{c^L_1}_{(\boldsymbol{x},\boldsymbol{\theta})}, \ldots, y^{c^L_M}_{(\boldsymbol{x},\boldsymbol{\theta})})^\top,$$

are the counterparts of noisy observations, where $y^{c^U_n}_{(\boldsymbol{x},\boldsymbol{\theta})} := c^U_n(\boldsymbol{x}, \boldsymbol{\theta}) + \epsilon^{c^U_n}$, $\epsilon^{c^U_n} \sim \mathcal{N}(0, \{\sigma^{c^U_n}_{\text{noise}}\}^2)$ and $y^{c^L_m}_{(\boldsymbol{x},\boldsymbol{\theta})} := c^L_m(\boldsymbol{x}, \boldsymbol{\theta}) + \epsilon^{c^L_m}$, $\epsilon^{c^L_m} \sim \mathcal{N}(0, \{\sigma^{c^L_m}_{\text{noise}}\}^2)$. We observe $(\boldsymbol{y}^f_{\boldsymbol{x},\boldsymbol{\theta}}, \boldsymbol{y}^g_{\boldsymbol{x},\boldsymbol{\theta}})$ at every BO iteration for selected $(\boldsymbol{x}, \boldsymbol{\theta})$, i.e., $\mathcal{D}_t = \{(\boldsymbol{x}_i, \boldsymbol{\theta}_i, \boldsymbol{y}^f_{\boldsymbol{x}_i,\boldsymbol{\theta}_i}, \boldsymbol{y}^g_{\boldsymbol{x}_i,\boldsymbol{\theta}_i})\}_{i=1}^n$ in the constraint setting. In addition to $f$ and $g$, the independent GPs are also fitted to $c^U_n$ and $c^L_m$, for which the posteriors given $\mathcal{D}_t$ are written as $\mathcal{N}(\mu^{c^U_n}_t(\boldsymbol{x}, \boldsymbol{\theta}), \{\sigma^{c^U_n}_t(\boldsymbol{x}, \boldsymbol{\theta})\}^2)$ and $\mathcal{N}(\mu^{c^L_m}_t(\boldsymbol{x}, \boldsymbol{\theta}), \{\sigma^{c^L_m}_t(\boldsymbol{x}, \boldsymbol{\theta})\}^2)$, respectively.

### E.1 Lower Bound

The MI and its lower bound can be derived by the same approach as (3):

$$\text{MI}(\boldsymbol{y}^f_{(\boldsymbol{x},\boldsymbol{\theta})}, \boldsymbol{y}^g_{(\boldsymbol{x},\boldsymbol{\theta})} ; f^*, g^*, \boldsymbol{x}^*, \boldsymbol{\theta}^* \mid \mathcal{D}_t) = \mathbb{E}_\Omega \left[ \log \frac{p(\boldsymbol{y}^f_{(\boldsymbol{x},\boldsymbol{\theta})}, \boldsymbol{y}^g_{(\boldsymbol{x},\boldsymbol{\theta})} \mid f^*, g^*, \boldsymbol{x}^*, \boldsymbol{\theta}^*, \mathcal{D}_t)}{p(\boldsymbol{y}^f_{(\boldsymbol{x},\boldsymbol{\theta})}, \boldsymbol{y}^g_{(\boldsymbol{x},\boldsymbol{\theta})} \mid \mathcal{D}_t)} \right]$$

$$= \mathbb{E}_{f^*, g^*, \boldsymbol{x}^*, \boldsymbol{\theta}^*} \left[ \mathbb{E}_{\boldsymbol{y}^f_{(\boldsymbol{x},\boldsymbol{\theta})}, \boldsymbol{y}^g_{(\boldsymbol{x},\boldsymbol{\theta})} \mid f^*, g^*, \boldsymbol{x}^*, \boldsymbol{\theta}^*, \mathcal{D}_t} \left[ \log \frac{q(\boldsymbol{y}^f_{(\boldsymbol{x},\boldsymbol{\theta})}, \boldsymbol{y}^g_{(\boldsymbol{x},\boldsymbol{\theta})} \mid f^*, g^*, \boldsymbol{x}^*, \boldsymbol{\theta}^*, \mathcal{D}_t)}{p(\boldsymbol{y}^f_{(\boldsymbol{x},\boldsymbol{\theta})}, \boldsymbol{y}^g_{(\boldsymbol{x},\boldsymbol{\theta})} \mid \mathcal{D}_t)} \right] \right.$$

$$\left. + \text{KL} \left( p(\boldsymbol{y}^f_{(\boldsymbol{x},\boldsymbol{\theta})}, \boldsymbol{y}^g_{(\boldsymbol{x},\boldsymbol{\theta})} \mid f^*, g^*, \boldsymbol{x}^*, \boldsymbol{\theta}^*, \mathcal{D}_t) \parallel q(\boldsymbol{y}^f_{(\boldsymbol{x},\boldsymbol{\theta})}, \boldsymbol{y}^g_{(\boldsymbol{x},\boldsymbol{\theta})} \mid f^*, g^*, \boldsymbol{x}^*, \boldsymbol{\theta}^*, \mathcal{D}_t) ) \right] $$

$$\geq \mathbb{E}_\Omega \left[ \log \frac{q(\boldsymbol{y}^f_{(\boldsymbol{x},\boldsymbol{\theta})}, \boldsymbol{y}^g_{(\boldsymbol{x},\boldsymbol{\theta})} \mid f^*, g^*, \boldsymbol{x}^*, \boldsymbol{\theta}^*, \mathcal{D}_t)}{p(\boldsymbol{y}^f_{(\boldsymbol{x},\boldsymbol{\theta})}, \boldsymbol{y}^g_{(\boldsymbol{x},\boldsymbol{\theta})} \mid \mathcal{D}_t)} \right] =: \text{LB}_c(\boldsymbol{x}, \boldsymbol{\theta}),$$

where here $\mathcal{D}_t^+ := \mathcal{D}_t \cup \{(\boldsymbol{x}^*, \boldsymbol{\theta}^*, f^*, g^*)\}$.

To define the variational distribution $q$, we follow the same approach as information-theoretic constraint BO proposed by (Takeno et al., 2022b). Let $\mathcal{A}^f = \{(c_0, \boldsymbol{c}) \mid c_0 \geq f^*, \boldsymbol{c} \geq 0, c_0 \in \mathbb{R}, \boldsymbol{c} \in \mathbb{R}^N\}$ and $\mathcal{A}^g = \{(c_0, \boldsymbol{c}) \mid c_0 \geq g^*, \boldsymbol{c} \geq 0, c_0 \in \mathbb{R}, \boldsymbol{c} \in \mathbb{R}^M\}$. When $f^*$ is given, $\mathcal{A}^f$ is the region that $\boldsymbol{h}^f_{\boldsymbol{x},\boldsymbol{\theta}^*(\boldsymbol{x})}$ cannot exist for $\forall \boldsymbol{x}$. When $g^*$ is given, $\mathcal{A}^g$ is the region that $\boldsymbol{h}^g_{\boldsymbol{x}^*,\boldsymbol{\theta}}$ cannot exist for $\forall \boldsymbol{\theta}$. Based on the same simplification of the conditioning discussed in section 3.1, we define the variational distribution as

$$q(\boldsymbol{y}^f_{(\boldsymbol{x},\boldsymbol{\theta})}, \boldsymbol{y}^g_{(\boldsymbol{x},\boldsymbol{\theta})} \mid f^*, g^*, \boldsymbol{x}^*, \boldsymbol{\theta}^*, \mathcal{D}_t) := p(\boldsymbol{y}^f_{(\boldsymbol{x},\boldsymbol{\theta})}, \boldsymbol{y}^g_{(\boldsymbol{x},\boldsymbol{\theta})} \mid \boldsymbol{h}^f_{(\boldsymbol{x},\boldsymbol{\theta}^*(\boldsymbol{x}))} \in \bar{\mathcal{A}}^f, \boldsymbol{h}^g_{(\boldsymbol{x}^*,\boldsymbol{\theta})} \in \bar{\mathcal{A}}^g, \mathcal{D}_t^+),$$

where $\bar{\mathcal{A}}^f$ and $\bar{\mathcal{A}}^g$ are the complement sets of $\mathcal{A}^f$ and $\mathcal{A}^g$, respectively. As a result, we see

$$\text{LB}_c(\boldsymbol{x}, \boldsymbol{\theta}) = \mathbb{E}_\Omega \left[ \log \frac{p(\boldsymbol{y}^f_{(\boldsymbol{x},\boldsymbol{\theta})}, \boldsymbol{y}^g_{(\boldsymbol{x},\boldsymbol{\theta})} \mid \boldsymbol{h}^f_{(\boldsymbol{x},\boldsymbol{\theta}^*(\boldsymbol{x}))} \in \bar{\mathcal{A}}^f, \boldsymbol{h}^g_{(\boldsymbol{x}^*,\boldsymbol{\theta})} \in \bar{\mathcal{A}}^g, \mathcal{D}_t^+)}{p(\boldsymbol{y}^f_{(\boldsymbol{x},\boldsymbol{\theta})}, \boldsymbol{y}^g_{(\boldsymbol{x},\boldsymbol{\theta})} \mid \mathcal{D}_t)} \right]$$

$$= \mathbb{E}_\Omega \left[ \log \frac{p(\boldsymbol{y}^f_{(\boldsymbol{x},\boldsymbol{\theta})} \mid \boldsymbol{h}^f_{(\boldsymbol{x},\boldsymbol{\theta}^*(\boldsymbol{x}))} \in \bar{\mathcal{A}}^f, \mathcal{D}_t^+)}{p(\boldsymbol{y}^f_{(\boldsymbol{x},\boldsymbol{\theta})} \mid \mathcal{D}_t)} + \log \frac{p(\boldsymbol{y}^g_{(\boldsymbol{x},\boldsymbol{\theta})} \mid \boldsymbol{h}^g_{(\boldsymbol{x}^*,\boldsymbol{\theta})} \in \bar{\mathcal{A}}^g, \mathcal{D}_t^+)}{p(\boldsymbol{y}^g_{(\boldsymbol{x},\boldsymbol{\theta})} \mid \mathcal{D}_t)} \right].$$

### E.2 Analytical Representation of Variational Distribution

From Bayes theorem,

$$p(\boldsymbol{y}^f_{(\boldsymbol{x},\boldsymbol{\theta})} \mid \boldsymbol{h}^f_{(\boldsymbol{x},\boldsymbol{\theta}^*(\boldsymbol{x}))} \in \bar{\mathcal{A}}^f, \mathcal{D}_t^+) = \frac{p(\boldsymbol{h}^f_{(\boldsymbol{x},\boldsymbol{\theta}^*(\boldsymbol{x}))} \in \bar{\mathcal{A}}^f \mid \boldsymbol{y}^f_{(\boldsymbol{x},\boldsymbol{\theta})}, \mathcal{D}_t^+) p(\boldsymbol{y}^f_{(\boldsymbol{x},\boldsymbol{\theta})} \mid \mathcal{D}_t^+)}{p(\boldsymbol{h}^f_{(\boldsymbol{x},\boldsymbol{\theta}^*(\boldsymbol{x}))} \in \bar{\mathcal{A}}^f \mid \mathcal{D}_t^+)}. \tag{22}$$

The density $p(\boldsymbol{h}^f_{(\boldsymbol{x},\boldsymbol{\theta}^*(\boldsymbol{x}))} \mid \boldsymbol{y}^f_{(\boldsymbol{x},\boldsymbol{\theta})}, \mathcal{D}_t^+)$ is an $(N+1)$-dimensional independent Gaussian distribution, for which the first dimension is $\mathcal{N}(m_1^f, \{s_1^f\}^2)$ shown in Appendix A.1 and from the second to the $(N+1)$-th dimension is $\mathcal{N}(m^{c_n^U}, \{s^{c_n^U}\}^2)$ where

$$m^{c_n^U} = \mu_t^{c_n^U}(\boldsymbol{x}, \boldsymbol{\theta}^*(\boldsymbol{x})) + \frac{\text{Cov}_t^{c_n^U}((\boldsymbol{x}, \boldsymbol{\theta}^*(\boldsymbol{x})), (\boldsymbol{x}, \boldsymbol{\theta}))}{\left\{ \sigma_t^{c_n^U}(\boldsymbol{x}, \boldsymbol{\theta}) \right\}^2} (y^{c_n^U}_{(\boldsymbol{x},\boldsymbol{\theta})} - \mu_t^{c_n^U}(\boldsymbol{x}, \boldsymbol{\theta})),$$

$$\{s^{c_n^U}\}^2 = \sigma_t^{c_n^U}(\boldsymbol{x}, \boldsymbol{\theta}^*(\boldsymbol{x})) - \frac{\left\{ \text{Cov}_t^{c_n^U}((\boldsymbol{x}, \boldsymbol{\theta}^*(\boldsymbol{x})), (\boldsymbol{x}, \boldsymbol{\theta})) \right\}^2}{\left\{ \sigma_t^{c_n^U}(\boldsymbol{x}, \boldsymbol{\theta}) \right\}^2}.$$

As a result, we can derive

$$p(\boldsymbol{h}^f_{(\boldsymbol{x},\boldsymbol{\theta}^*(\boldsymbol{x}))} \in \bar{\mathcal{A}}^f \mid \boldsymbol{y}^f_{(\boldsymbol{x},\boldsymbol{\theta})}, \mathcal{D}^+_t) = 1 - (1 - \Phi(\frac{f^* - m^f_1}{s^f_1})) \prod_{n=1}^{N} (1 - \Phi(\frac{0 - m^{c^U_n}}{s^{c^U_n}})),$$

$$p(\boldsymbol{h}^f_{(\boldsymbol{x},\boldsymbol{\theta}^*(\boldsymbol{x}))} \in \bar{\mathcal{A}}^f \mid \mathcal{D}^+_t) = 1 - (1 - \Phi(\frac{f^* - m^f_2}{s^f_2})) \prod_{n=1}^{N} (1 - \Phi(\frac{0 - \mu^{c^U_n}_t(\boldsymbol{x},\boldsymbol{\theta}^*(\boldsymbol{x}))}{\sigma^{c^U_n}_t(\boldsymbol{x},\boldsymbol{\theta}^*(\boldsymbol{x}))})),$$

$$p(\boldsymbol{y}^f_{(\boldsymbol{x},\boldsymbol{\theta})} \mid \mathcal{D}^+_t) = \phi(\frac{y^f_{(\boldsymbol{x},\boldsymbol{\theta})} - m^f_3}{s^f_3}) \prod_{n=1}^{N} \phi(\frac{y^{c^U_n}_{(\boldsymbol{x},\boldsymbol{\theta})} - \mu^{c^U_n}_t(\boldsymbol{x},\boldsymbol{\theta})}{\sigma^{c^U_n}_t(\boldsymbol{x},\boldsymbol{\theta})})/(s^f_3\,\sigma^{c^U_n}_t(\boldsymbol{x},\boldsymbol{\theta})).$$

Similarly, for the lower-level density, Bayes theorem transforms

$$p(\boldsymbol{y}^g_{(\boldsymbol{x},\boldsymbol{\theta})} \mid \boldsymbol{h}^g_{(\boldsymbol{x}^*,\boldsymbol{\theta})} \in \bar{\mathcal{A}}^g, \mathcal{D}^+_t) = \frac{p(\boldsymbol{h}^g_{(\boldsymbol{x}^*,\boldsymbol{\theta})} \in \bar{\mathcal{A}}^g \mid \boldsymbol{y}^g_{(\boldsymbol{x},\boldsymbol{\theta})}, \mathcal{D}^+_t) p(\boldsymbol{y}^g_{(\boldsymbol{x},\boldsymbol{\theta})} \mid \mathcal{D}^+_t)}{p(\boldsymbol{h}^g_{(\boldsymbol{x}^*,\boldsymbol{\theta})} \in \bar{\mathcal{A}}^g \mid \mathcal{D}^+_t)}$$

Here again, the density of the first dimension of $p(\boldsymbol{h}^g_{(\boldsymbol{x}^*,\boldsymbol{\theta})} \mid \boldsymbol{y}^g_{(\boldsymbol{x},\boldsymbol{\theta})}, \mathcal{D}^+_t)$ is $\mathcal{N}(m^g_1, \{s^g_1\}^2)$ shown in Appendix A.1 and from the second to the $(M+1)$-th dimension is $\mathcal{N}(m^{c^L_m}, \{s^{c^L_m}\}^2)$ where

$$m^{c^L_m} = \mu^{c^L_m}_t(\boldsymbol{x}^*,\boldsymbol{\theta}) + \frac{\text{Cov}^{c^L_m}_t((\boldsymbol{x}^*,\boldsymbol{\theta}),(\boldsymbol{x},\boldsymbol{\theta}))}{\left\{\sigma^{c^L_m}_t(\boldsymbol{x},\boldsymbol{\theta})\right\}^2}(y^{c^L_m}(\boldsymbol{x},\boldsymbol{\theta}) - \mu^{c^L_m}_t(\boldsymbol{x},\boldsymbol{\theta})),$$

$$\{s^{c^L_m}\}^2 = \sigma^{c^L_m}_t(\boldsymbol{x}^*,\boldsymbol{\theta}) - \frac{\left\{\text{Cov}^{c^L_m}_t((\boldsymbol{x}^*,\boldsymbol{\theta}),(\boldsymbol{x},\boldsymbol{\theta}))\right\}^2}{\left\{\sigma^{c^L_m}_t(\boldsymbol{x},\boldsymbol{\theta})\right\}^2}.$$

As a result

$$p(\boldsymbol{h}^g_{(\boldsymbol{x}^*,\boldsymbol{\theta})} \in \bar{\mathcal{A}}^g \mid \boldsymbol{y}^g_{(\boldsymbol{x},\boldsymbol{\theta})}, \mathcal{D}^+_t) = 1 - \left(1 - \Phi\left(\frac{g^* - m^g_1}{s^g_1}\right)\right) \prod_{m=1}^{M} \left(1 - \Phi\left(\frac{0 - m^{c^L_m}}{s^{c^L_m}}\right)\right)$$

$$p(\boldsymbol{h}^g_{(\boldsymbol{x}^*,\boldsymbol{\theta})} \in \bar{\mathcal{A}}^g \mid \mathcal{D}^+_t) = 1 - \left(1 - \Phi\left(\frac{g^* - m^g_2}{s^g_2}\right)\right) \prod_{m=1}^{M} \left(1 - \Phi\left(\frac{0 - \mu^{c^L_m}_t(\boldsymbol{x}^*,\boldsymbol{\theta})}{\sigma^{c^L_m}_t(\boldsymbol{x}^*,\boldsymbol{\theta})}\right)\right)$$

$$p(\boldsymbol{y}^g_{(\boldsymbol{x},\boldsymbol{\theta})} \mid \mathcal{D}^+_t) = \phi\left(\frac{y^g_{(\boldsymbol{x},\boldsymbol{\theta})} - m^g_3}{s^g_3}\right) \prod_{m=1}^{M} \phi\left(\frac{y^{c^L_m}_{(\boldsymbol{x},\boldsymbol{\theta})} - \mu^{c^L_m}_t(\boldsymbol{x},\boldsymbol{\theta})}{\sigma^{c^L_m}_t(\boldsymbol{x},\boldsymbol{\theta})}\right)/(s^g_3\,\sigma^{c^L_m}_t(\boldsymbol{x},\boldsymbol{\theta}))$$

# F SUPPLEMENTARY FOR EXPERIMENTS

## F.1 OTHER DETAILS OF EXPERIMENTAL SETTINGS

We used the `SingleTaskGP` model of BoTorch (Balandat et al., 2020) to define the GPs. The output of each benchmark function are transformed by signed log1p function $\text{sign}(y)\log(1+|y|)$ except for BraninHoo and Goldstein-price for which the transformation shown by (Picheny et al., 2013) was used. For benchmark functions, the input space is scaled to $[0,1]^{d_X+d_\Theta}$ from the original input domain. For the chemical dataset, we re-defined the true objective function values as $f(\boldsymbol{x},\boldsymbol{\theta}) = -\log((\max_{\boldsymbol{x}',\boldsymbol{\theta}'} f_{\text{ori}}(\boldsymbol{x}',\boldsymbol{\theta}') + 10^{-4}) - f_{\text{ori}}(\boldsymbol{x},\boldsymbol{\theta}))$ and $g(\boldsymbol{x},\boldsymbol{\theta}) = -\log((\max_{\boldsymbol{x}',\boldsymbol{\theta}'} g_{\text{ori}}(\boldsymbol{x}',\boldsymbol{\theta}') + 10^{-4}) - g_{\text{ori}}(\boldsymbol{x},\boldsymbol{\theta}))$, where $f_{\text{ori}}$ and $g_{\text{ori}}$ are original functions in the dataset. This is a log transformation applied to the difference from the maximum value (instead of 0). We employed this transformation because, in this dataset, many values are concentrated on the small scale differences around $\max f_{\text{ori}}(\boldsymbol{x},\boldsymbol{\theta})$ and $\max g_{\text{ori}}(\boldsymbol{x},\boldsymbol{\theta})$, respectively.

## F.2 LARGER NOISE SETTING

Figure 7 shows results with a stronger noise setting (the noise standard deviation is set as $10^{-1}$). We do not see large difference for the relative performance among compared methods compared with the small noise setting shown in Fig. 4.

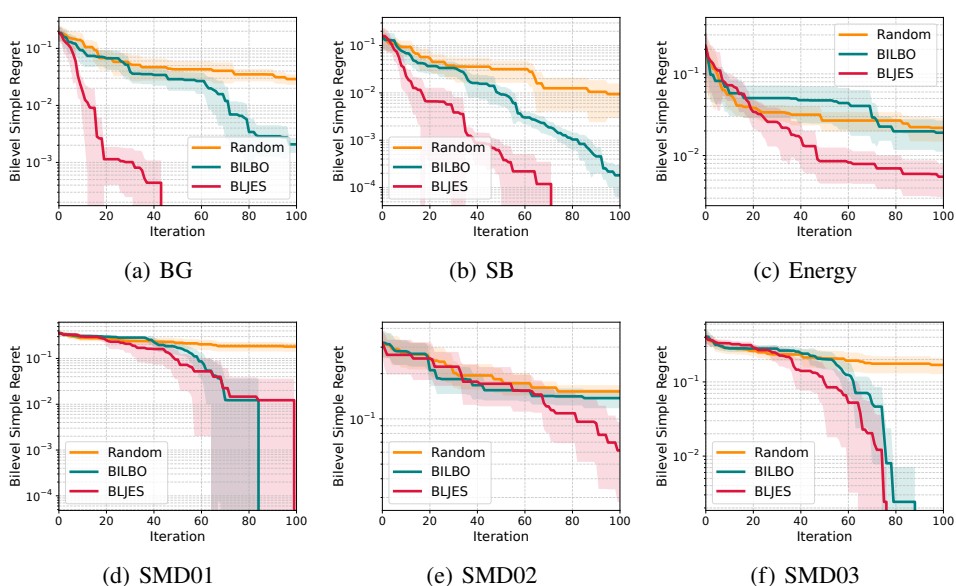

(a) BG  (b) SB  (c) Energy

(d) SMD01  (e) SMD02  (f) SMD03

Figure 7: Regret comparison with $10^{-1}$ noise standard deviation.

### F.3 ADDITIONAL RESULTS ON DECOUPLED SETTING

Figure 8 shows regret in decoupled setting for all different length scale settings of the GP prior, which generates true objectives. We obviously see that BLJES was superior or comparable to BILBO.

### F.4 ADDITIONAL RESULTS ON EFFECT OF THE NUMBER OF SAMPLINGS

Figure 9 shows results of BLJES for different $K$. We do not see particularly large differences among different $K$ settings in these benchmarks.

### F.5 CONTINUOUS DOMAIN

Figure 10 shows the regret in the case of $\mathcal{X}$ and $\Theta$ are the continuous space. We employed gradient based optimizers for both of the bilevel problem defined by sample paths and the acquisition function maximization (gradient of a bilevel problem is discussed in Appendix C). Here, BILBO is not performed because Chew et al. (2025) only discuss the finite domain. We see that BLJES efficiently decreases the regret even in the continuous space. Only in SMD02, BLJES was not efficient compared with the random selection.

### F.6 CONSTRAINT PROBLEMS

For empirical evaluation, we employed problems from the bilevel optimization benchmark (Sinha et al., 2014), denoted as SDM09 ($d_{\mathcal{X}} = 2, d_{\Theta} = 2, N = 1, M = 1$), 10 ($d_{\mathcal{X}} = 2, d_{\Theta} = 2, N = 2, M = 1$), 11 ($d_{\mathcal{X}} = 2, d_{\Theta} = 2, N = 1, M = 1$), and 12 ($d_{\mathcal{X}} = 2, d_{\Theta} = 2, N = 3, M = 2$). The number of grid points in each dimension is 10 ($10^4$ points). The evaluation metric is

$$\min_{i \in [n_0 + t]} \max_{h \in \{f, g, c_1^U, \dots, c_N^U, c_1^L, \dots, c_M^L\}} r_h(\boldsymbol{x}_i, \boldsymbol{\theta}_i)$$

where $r_h$ become $r_f$ and $r_g$ shown in section 5 if $h = f$ or $g$, and

$$r_c(\boldsymbol{x}_i, \boldsymbol{\theta}_i) = \max(0, -c(\boldsymbol{x}_i, \boldsymbol{\theta}_i)) / \max_{\boldsymbol{x}, \boldsymbol{\theta}}(\max(0, -c(\boldsymbol{x}, \boldsymbol{\theta}))), c \in \{c_1^U, \dots, c_N^U, c_1^L, \dots, c_M^L\}$$

if $h \in \{c_1^U, \dots, c_N^U, c_1^L, \dots, c_M^L\}$. The other settings are same as described in the beginning of section 5.

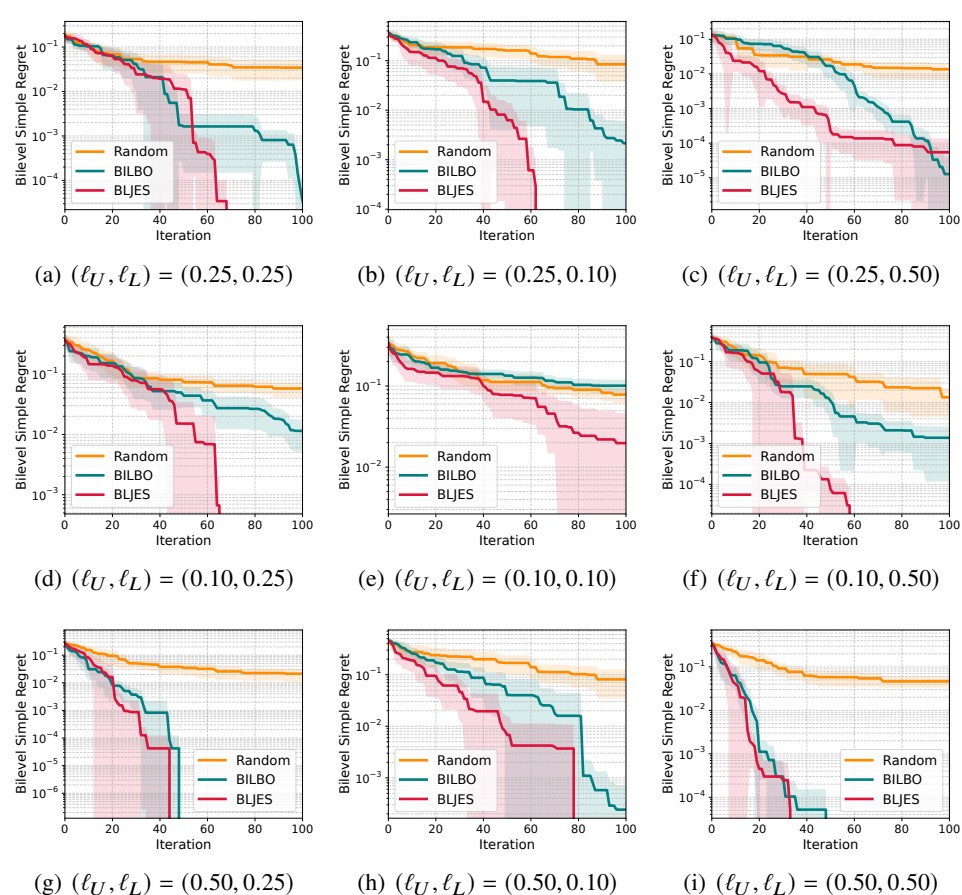

$$(a) \ (\ell_U, \ell_L) = (0.25, 0.25) \quad (b) \ (\ell_U, \ell_L) = (0.25, 0.10) \quad (c) \ (\ell_U, \ell_L) = (0.25, 0.50)$$

$$(d) \ (\ell_U, \ell_L) = (0.10, 0.25) \quad (e) \ (\ell_U, \ell_L) = (0.10, 0.10) \quad (f) \ (\ell_U, \ell_L) = (0.10, 0.50)$$

$$(g) \ (\ell_U, \ell_L) = (0.50, 0.25) \quad (h) \ (\ell_U, \ell_L) = (0.50, 0.10) \quad (i) \ (\ell_U, \ell_L) = (0.50, 0.50)$$

Figure 8: Regret comparison on functions from the GP prior under decoupled setting.

The results are in Fig. 11. For SMD09 and SMD11, BLJES shows faster decrease of the regret. For SMD12, BLJES and BILBO are comparable and both of them are much better than Random. For SMD10, BLJES rapidly decreased the regret, while BLJES also quickly decreased the regret (the difference is in small scale values).

### F.7 HIGHER DIMENSIONAL PROBLEMS

By using the GP prior function, we evaluate performance on higher dimensional problems. Figure 12 shows the results on $d_{\mathcal{X}} = d_{\Theta} = 4$ and $d_{\mathcal{X}} = d_{\Theta} = 5$. Since creating fine grid points is difficult for these settings, the query setting (continuous domain) is employed here, because of which BILBO is not shown. We see that BLJES shows reasonable performance on the four dimension problems (a)-(d), while performance difference from the random selection becomes unclear on the five dimension problems (a)-(d).

In general, it is widely known that higher dimensional problems (e.g., more than 10 dimension) are difficult for BO (e.g., Santoni et al., 2024). In our bilevel problem, the dimension of the search space is $d_{\mathcal{X}} + d_{\Theta}$, while the surrogate model is estimated on $d_{\mathcal{X}}$ and $d_{\Theta}$ spaces separately. We conjecture that our results are consistent with the general consensus of the high dimensional performance of BO by considering the additional difficulty caused by the low-level problem optimality constraint. Many studies exist for dealing with high dimensional problems (e.g., reviewer by Malu et al., 2021; González-Duque et al., 2024), but the efficient high dimensional exploration is still an important open problem in the context of BO. Some existing strategies for high dimensional problems are applicable to BLJES. For example, the well-known random projection-based method called REMBO (Wang et al., 2016) is applicable just by preparing two random projections for $x$ and $\theta$. Another

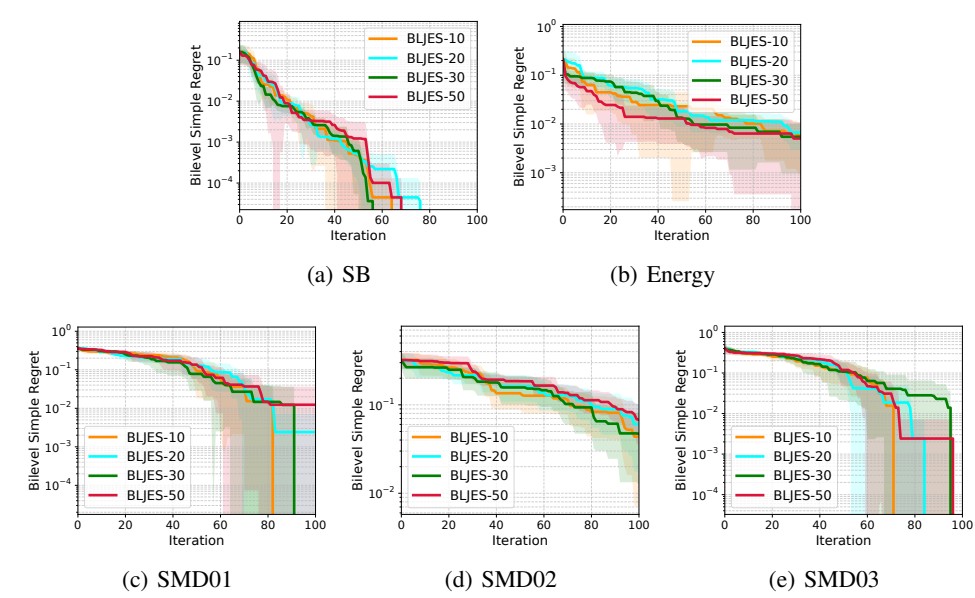

Figure 9: BLJES with different $K$.

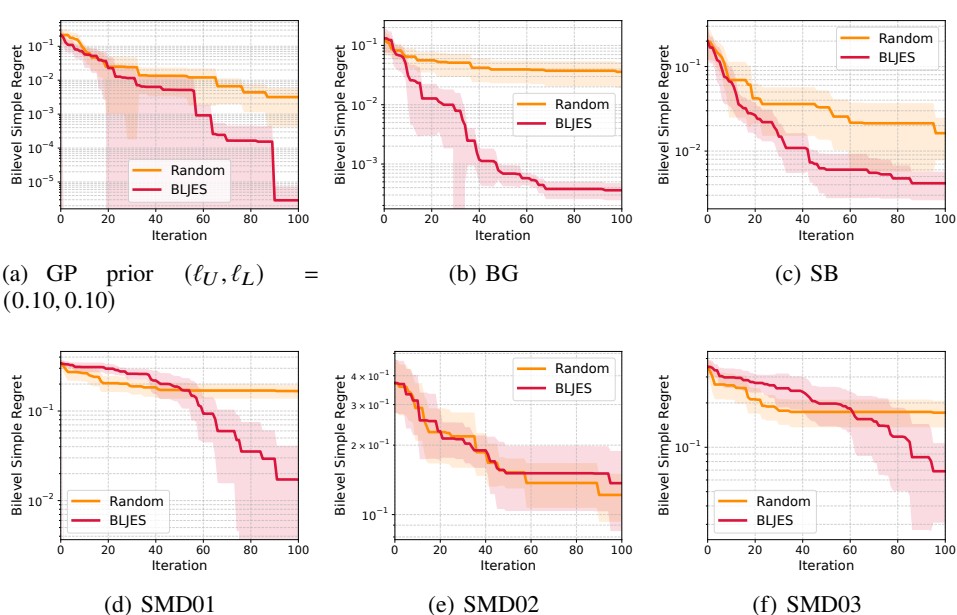

Figure 10: Regret comparison on continuous input domain.

well-known general strategy is LineBO (Kirschner et al., 2019) that explores one-dimensional (or low-dimensional) subspace at each iteration. In the case of BLJES, by defining one-dimensional subspace for each of $x$ and $\theta$, the same procedure as LineBO can be performed. However, detailed investigation for high dimensional setting is out of scope of this paper and it should be an important future direction.

## F.8 BLJES WITHOUT TRUNCATION CONDITIONS

From the BLJES criterion (5), we consider a variant that removes the truncation conditions $f(x, \theta^*(x)) \leq f^*$ and $g(x^*, \theta) \leq g^*$ as shown in (21). From the viewpoint of the independence

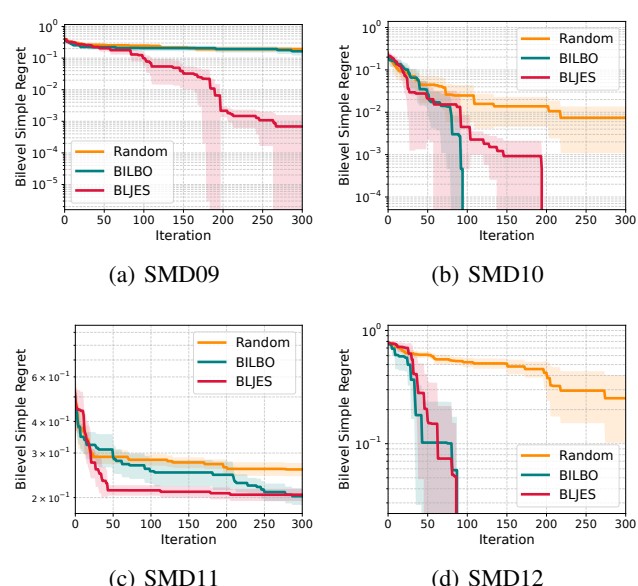

(a) SMD09  (b) SMD10

(c) SMD11  (d) SMD12

Figure 11: Regret comparison in constraint problems.

approximation, 'BLJES without truncations' can be seen as a variational lower bound with a stronger independence assumption as described in Appendix B.2.

The results are shown in Fig. 13. We can clearly see that 'w/o truncation' is much worse than 'w/ truncation'. This indicates that the truncation conditions largely contributes to represent effectiveness of each candidate point, which is consistent with our discussion in Appendix B.

### F.9 Effect of Random Fourier Approximation

We use RFF for sampling $\Omega$ in the BLJES computation as described in section 3.2. In the case of the pool setting (finite domain), the sampling from the original GPs of $f$ and $g$ is also possible as far as the sizes of $|\mathcal{X}|$ and $|\Theta|$ are moderate (because direct implementation of the sampling from the GP posterior requires $O(|\mathcal{X}|^3)$ and $O(|\Theta|^3)$). Therefore, based on the pool setting that is used in section 5, we here evaluate performance difference between BLJES with the original GP posterior sampling and that with the RFF-based sampling. The results on the BG benchmark are in Fig. 14. We see that the transition of the regret is highly similar, which suggests that RFF did not have large effect on the performance in this dataset.

### F.10 Combined Error by MC approximation and Random Fourier Feature

We here evaluate the quality of computations to approximate the true lower bound (5). As described in section 3.2, we use the MC approximation for the expectation $\mathbb{E}_\Omega$, and RFF is also used for sampling the optimal solutions and values. The pseudo ground truth of (5) was created by using the sampling from the original GP posteriors of $f$ and $g$ with the number of samplings $K = 10^4$. The mean squared error compared with this pseudo ground truth is shown in Fig. 15 (10 trials). 'Exact' indicates the GP posterior, and RFF-1000 and RFF-500 are RFF $D = 1000$ and $D = 500$, respectively. Overall, 'Exact' shows lower errors, while the error of RFF decreased for the larger $D$. When $K$ increases, the error of RFF converges to some non-zero value, which approximately represents the error purely caused by RFF. We see that the error rapidly decreased with the increase of $K$, which suggests fast convergence of the MC sampling. As a results, we do not see a severe deterioration by the combination of the MC approximation and RFF.

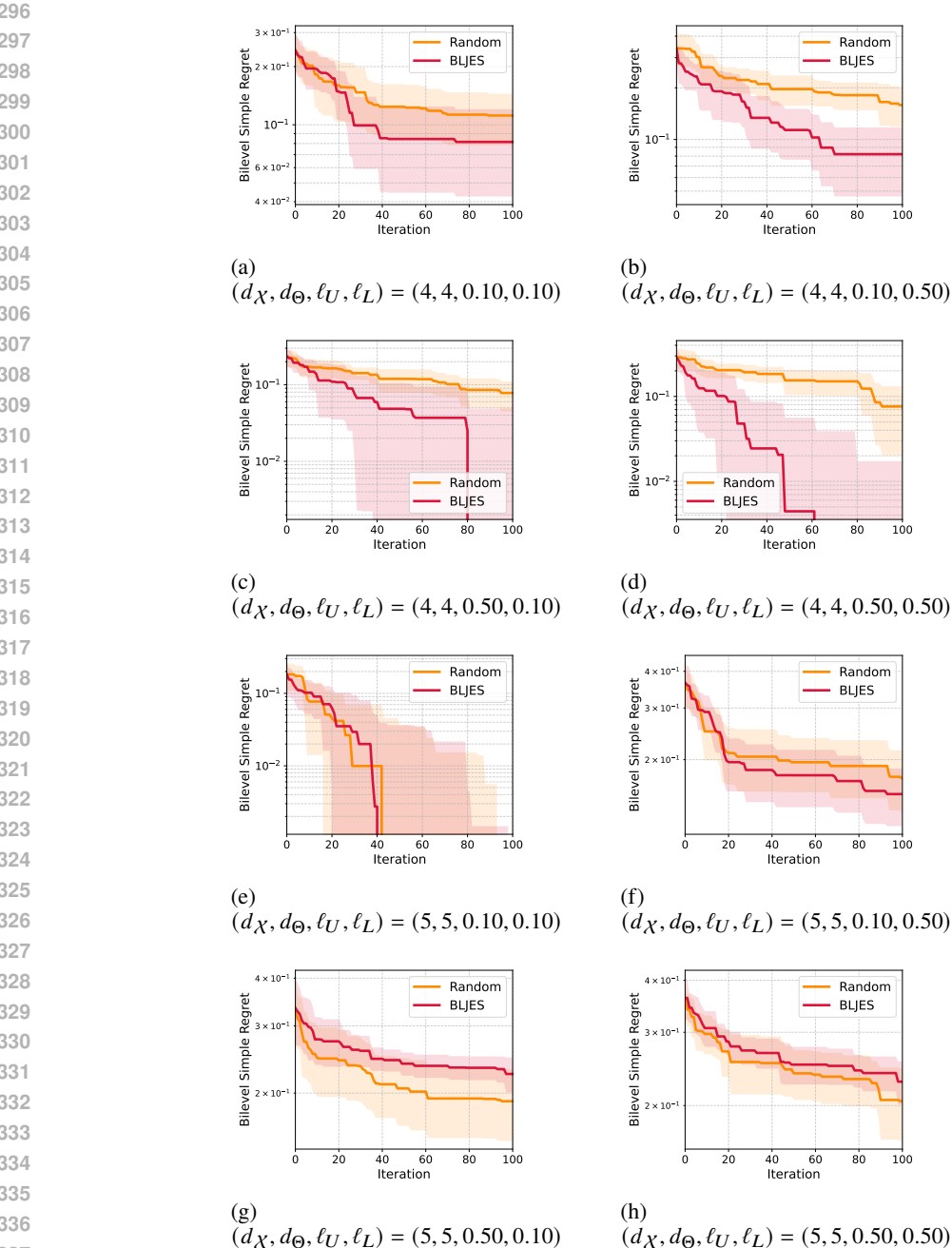

Figure 12: Regret comparison on functions from the GP prior.

# G    EXISTING STUDIES FOR REGRET ANALYSIS OF INFORMATION-THEORETIC BO

The well-known max-value entropy search (MES) (Wang & Jegelka, 2017) provides the regret bound for the special case in which only one Monte-Carlo (MC) sample is used for its expectation approximation. However, technical problems in their proof were pointed out by (Takeno et al., 2022b). Takeno et al. (2024) provided the regret bound of an acquisition function equivalent to the one sample MES, but it is still for the one sample special case that cannot be applied to the usual MC approximation by multiple samples. In the context of information-theoretic BO for (single-level) multi-objective optimization, Belakaria et al. (2019) discussed a regret bound, but Suzuki et al. (2020) pointed out an obvious significant mistake.

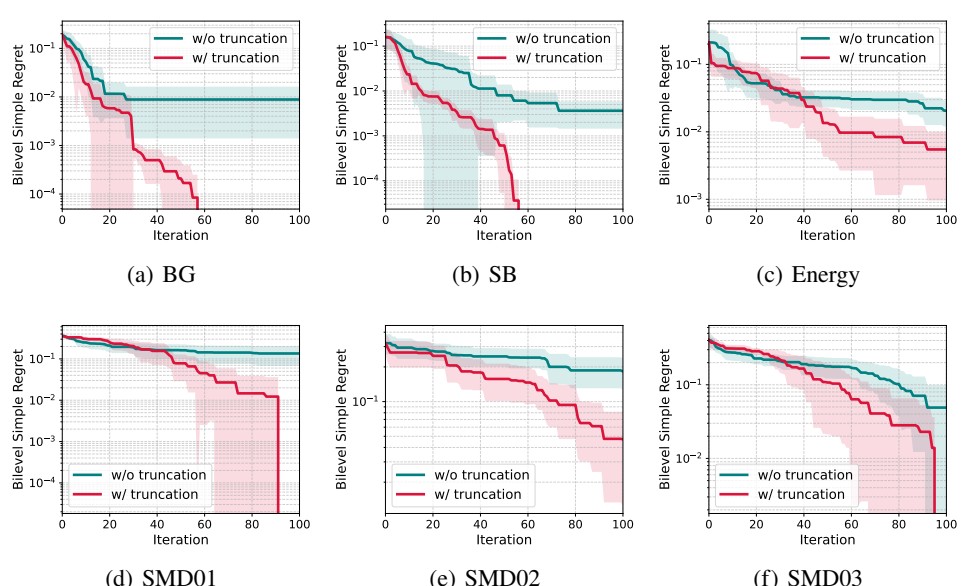

Figure 13: Evaluation of truncation in BLJES on benchmark problems.

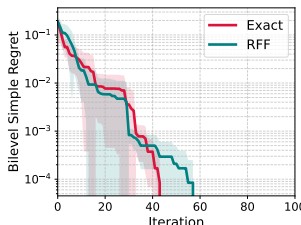

Figure 14: Evaluation of BLJES with the original GP posterior sampling and with RFF-based sampling (BG benchmark).

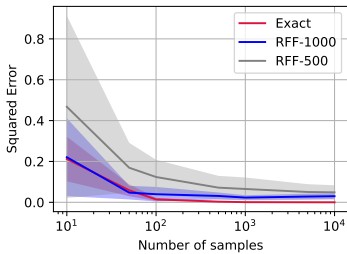

Figure 15: Approximation error of MC sampling and RFF (BG benchmark).

# H LLM USAGE

In this manuscript, LLM was only used to polish writing.

