# OpenReview forum: "Information-Theoretic Bayesian Optimization for Bilevel Optimization Problems"
_ICLR.cc/2026/Conference — Submitted to ICLR 2026_

### Official Review · Reviewer_nt7w · 2025-10-22

**Soundness:** 3
**Presentation:** 3
**Contribution:** 3
**Rating:** 6
**Confidence:** 4

**Summary:**

The paper proposes Bilevel Optimization via Lower-bound based Joint Entropy Search (BLJES), a bilevel Bayesian optimization method grounded in an information-theoretic framework.
BLJES simultaneously considers the information gain regarding both the upper-level and lower-level optima (points and values), thereby defining a unified acquisition criterion that evaluates the overall benefit across both levels.

Because exact maximization of this two-level information gain is computationally intractable, the authors derive a variational lower-bound approximation for it and show that this formulation can be viewed as a natural extension of single-level information-theoretic Bayesian optimization.

The effectiveness of the proposed method is empirically demonstrated through experiments using sample-path functions from Gaussian processes as well as several benchmark optimization functions.

**Strengths:**

While BILBO can only be applied to Bayesian optimization on discrete domains by design, this paper derives the gradient of the BLJES objective function, enabling its application to continuous domains as well. This constitutes a qualitative and practical advantage of BLJES over BILBO.
Also BILBO shows superior performance comparede to BILBO in the experimental results reported in the paper.

**Weaknesses:**

Compared with BILBO, the following weak points can be identified:

- Lack of theoretical performance guarantees.

- As I understand it, BLJES, when extended to constrained bilevel optimization, evaluates the information gain under feasibility conditions by approximating the ``predictive distribution given the optimal value'' with a truncated variational distribution, following the extension proposed by Takeno et al. (2022b).
In other words, during the evaluation of the acquisition function, the predictive distribution is truncated by the feasibility constraint. However, this paper does not provide theoretical guarantees on zero-violation assurance.
Although the experiments show good performance on constrained benchmarks, BLJES does not include a safety mechanism (e.g., trust region) that ensures all samples are taken strictly within the feasible set. It lacks an explicit guarantee for high-probability constraint satisfaction so some users would worry about using BLJES.

- It is better to provide source code and dataset for improving the reproducibility of the work.

**Questions:**

- In information-theoretic approaches, is it fundamentally impossible to provide some form of theoretical guarantee, such as a no-regret bound?
While the experiments presented in this paper convincingly demonstrate strong empirical performance, having a theoretical guarantee would make the method more trustworthy for practical use.

- I can also imagine applications where it is important to guarantee constraint satisfaction with high probability.
Is it possible to ensure this—similarly to BILBO—by combining BLJES with an additional trust-region–style feasibility filter or other operational safety mechanism as a front-end module, so that the method can satisfy constraints with high probability? If not, why?

---

> ### Author Response · Authors · 2025-12-03
>
> Thank you for your constructive comments. We summarize our answer as the following three subjects (reference can be found in our paper).
>
> 1.
>
> > regret bound
>
> See our response #1 to reviewer k4Ay.
>
> 2.
>
> > high-probability constraint satisfaction
>
> BILBO defines a penalty (regret) for the constraint violations, including violations on 1) all constraints of high- and low- problems, and 2) the optimality of the lower-level problem. Those violations can be bounded by using the posterior variance if the input is in sets that they call 1) trusted set of feasible solution and 2) trusted set of optimal lower-level solutions, respectively. This can be seen as the high-probability constraint satisfaction in BILBO.
>
> The two trusted sets are constructed based on the well-known probabilistic bound on the error of the posterior mean (e.g., Lemma 5.1 in Srinivas et al., ICML 2010). This bound does not depend on the acquisition function (the bound holds for arbitrary selections of the training points). Therefore, if the user wants the probabilistic guarantee at some iteration of BLJES, a simple strategy is to restrict the candidate $(\boldsymbol{x}, \boldsymbol{\theta})$ only inside the intersection of two trusted sets.
>
> 3.
>
> > code
>
> We uploaded our code as supplementary.

---

### Official Review · Reviewer_k4Ay · 2025-10-28

**Soundness:** 3
**Presentation:** 2
**Contribution:** 2
**Rating:** 4
**Confidence:** 4

**Summary:**

This paper addresses bilevel optimization problems where both the upper and lower level objective functions are expensive black-box functions.  The authors are the first to introduce information-theoretic principles to this domain, proposing a new acquisition function named BLJES (Bilevel optimization via Lower-bound based Joint Entropy Search). The core idea is to shift the optimization goal from directly seeking higher function values to maximizing the bilevel information gain regarding the optimal solutions $(x^\*, \theta^\*)$ and their corresponding optimal values $(f^\*, g^\*)$ for both levels. This unified decision criterion balances exploration across both levels of the problem. As direct computation of this information gain (which is the mutual information) is intractable, the paper derives a computable variational lower bound. A key technical contribution is the creative extension of the "truncation approximation" concept, which is mature in single-level BO, to the complex structure of bilevel problems. Furthermore, the paper demonstrates the flexibility of this information-theoretic framework, showing its natural extension to the decoupled setting where observations can be separated and to more complex scenarios involving expensive black-box constraints. Experimental results show that BLJES achieves competitive performance against the current state-of-the-art method, BILBO, on various synthetic datasets and benchmark problems.

**Strengths:**

- This work is the first to establish a rigorous information-theoretic framework for bilevel Bayesian optimization. The proposed bilevel information gain concept offers a novel perspective for tackling this complex problem.

- The paper successfully transforms a theoretically ideal yet computationally intractable objective (mutual information) into an engineering-feasible algorithm through a series of well-founded approximations, including the variational lower bound, truncation approximation, and Monte Carlo sampling. The extension of the truncation approximation from the single-level to the bilevel case appears intuitively sound.

**Weaknesses:**

- A significant shortcoming of this paper is the absence of theoretical analysis for the proposed algorithm. The authors note in the introduction that the SOTA comparison, BILBO, possesses a theoretical regret guarantee, yet this paper provides no equivalent theoretical support for BLJES.  Although the experimental section uses bilevel simple regret as an evaluation metric and demonstrates good empirical convergence, this is not a substitute for formal theoretical proof. The absence of theoretical results such as a regret bound leaves the algorithm’s sample complexity and convergence behavior under finite budgets uncharacterized.  This is a considerable theoretical gap for a Bayesian optimization method, where sample efficiency is central to its purpose.

- All experiments are conducted in extremely low-dimensional spaces (both upper and lower dimensions $d_X, d_\Theta \leq 2$).  This severely limits the generality of the results. It is well known that BO methods suffer from the curse of dimensionality, and the nested nature of bilevel optimization likely worsens this challenge.  Thus, success on low-dimensional synthetic problems does not demonstrate scalability or robustness of BLJES in realistic, high-dimensional cases.

-  Although one experiment includes a simulator-based energy market problem, the vast majority still rely on synthetic functions sampled from GP priors and standard academic benchmarks.  To convincingly demonstrate the method’s practical value, validation on real-world black-box problems (e.g., engineering design, hyperparameter tuning) is needed.  The current experimental setup feels thin and does not fully exhibit the method’s ability to handle complex, noisy, or constrained real-world tasks.

**Questions:**

The core methodology relies on a variational lower bound to approximate the intractable mutual information. The quality of this bound, and the validity of maximizing it as a proxy for the true information gain, critically depends on how well the chosen variational distribution $q$ approximates the true posterior $p$. The paper mainly justifies this approach by citing its empirical success in prior work, but does not provide a theoretical analysis of the bound's tightness in this more complex bilevel optimization setting.

Can the authors provide more theoretical results regarding the tightness of this bound? For example, is it possible to theoretically analyze the error induced by this approximation, or under what conditions this lower bound becomes a closer approximation to the true mutual information? Providing analysis in this area would significantly strengthen the theoretical completeness of the proposed method.

---

> ### Author Response · Authors · 2025-12-03
>
> Thank you for your constructive comments. We summarize our answer as the following four subjects (reference can be found in our paper).
>
> 1.
>
> > regret guarantee
>
> For general information-theoretic BO, the regret analysis is an well-known open problem even for the simplest single-level standard problem setting.
>
> Max-value entropy search (MES) (Wang & Jegelka, ICML 2017) provides the regret bound for the special case of MES with only one Monte-Carlo (MC) sample of its expectation approximation. However, a technical issue in their proof was pointed out by (Takeno et al., ICML 2022b). Takeno et al., ICML 2024 provided the regret bound of an acquisition function equivalent to the one sample MES, but it is still for the one sample special case that cannot be applied to the usual MC approximation by multiple samples. In the context of information-theoretic BO for (single-level) multi-objective optimization, Berakaria et al., NeurIPS 2020 discussed a regret bound, but Suzuki et al., ICML 2020 pointed out an obvious significant mistake.
>
> As a result, the regret bound of information-theoretic BO still has not been revealed even for the single-level setting. The regret bound has been widely studied particularly for GP-UCB based approaches including BILBO. However, constant factors are usually quite large and ignored in the discussion by which fast convergence of the bound does not immediately indicate practical performance. Further, the balance parameter, adjusting the effect of the posterior mean and variance, is often difficult to select (Theoretical suggestions are not necessarily good in practice. For example, the original GP-UCB paper (Srinivas et al., ICML 2010) manually adjusted this parameter).
>
> We added the discussion at the second paragraph of section 6.
>
> 2.
>
> > dimension
>
> The setting $d_{\cal X} = d_{\Theta} = 2$ has a four dimensional search space because the candidate space is the product space $\cal{X} \times \Theta$. Therefore, we do not consider this setting is 'extremely low-dimensional space' compared with the typical BO papers, in which problems around four dimension is common (Further, the bilevel setting should be more difficult than the standard single-level setting because of the optimality constraint of the lower-level problem).
>
> In general, even for the single-level problem, high dimensional problems are still not easy for most of BO methods though there have been many studies (e.g., reviewed by Malu et al., IISA 2021 and González-Duque et al., NeurIPS 2024). On the other hand, for (both-level expensive) bilevel BO, there is still no consensus even on how an acquisition function can be constructed unlike the single level setting (as reviewed in introduction, there are only a few papers for this setting), and we do not intend to extend the scope of our paper to high-dimensional settings. We added this discussion about high dimensional problems as one of important future direction in the first paragraph of section 6.
>
> Further, in the revised manuscript, we added $d_{\cal X} = d_{\Theta} = 4$ and $d_{\cal X} = d_{\Theta} = 5$ GP derived synthetic functions in Appendix F.7. We see that, for $d_{\cal X} = d_{\Theta} = 4$ (eight dimensional search space), BLJES worked reasonably, while for $d_{\cal X} = d_{\Theta} = 5$ (ten dimensional search space), performance becomes similar to the random search. We conjecture that these results are consistent with the empirical knowledge of BO observed in the community.
>
> 3.
>
> > real-world black-box problem
>
> We newly added the Chemical dataset in Section 5.2. See response #3 to reviewer pjE9 for detail.
>
> Note that the hyper-parameter (HP) tuning of machine learning algorithms is not a problem fully suitable for our problem setting. We assume that both level problems are expensive black-box problems. On the other hand, in HP tuning, the lower-level objective function (training loss) is usually white-box and not expensive (in other words, for the lower-level problem, BO is not used). About typical targets that we assume, see response #1 to reviewer pjE9.

---

> > ### Author Response · Authors · 2025-12-03
> >
> > 4.
> >
> > > analysis of the bound
> >
> > Our lower bound is based on a variational distribution (4) defined via three conditions described in the itemized list after (4). In Appendix B, we newly added an analysis about under what assumption this variational distribution can be justified. The consequence is that when we assume the independence on the posterior predictive distribution, our variational distribution is derived. In other words, under the conditional independence assumption, our lower bound becomes equal to the original MI. In this reply, we only briefly describe the main idea. See Appendix B for detail.
> >
> > Approximate computations based on independence assumptions are popular in the context of variational inference. A characteristics of our approach is that we preserve dependency between a candidate $(\boldsymbol{x}, \boldsymbol{\theta})$ and $(\boldsymbol{x},\boldsymbol{\theta}^{\*}(\boldsymbol{x}))$ for $f$, and $(\boldsymbol{x}, \boldsymbol{\theta})$ and $(\boldsymbol{x}^{\*},\boldsymbol{\theta})$ for $g$ instead of assuming full independence of $(\boldsymbol{x}, \boldsymbol{\theta})$ from other points. This derives truncation conditions $f(\boldsymbol{x}, \boldsymbol{\theta}^{\*}(\boldsymbol{x})) \leq f^{\*}$ and $g(\boldsymbol{x}^{\*}, \boldsymbol{\theta}) \leq g^{\*}$, which can be seen as an extension of the truncation conditions widely used in information-theoretic BO (e.g., Wang & Jegelka, ICML 2017; Takeno et al., Neural Comput. 2022).
> >
> > In existing information-theoretic BO methods, specific form of truncate distributions are directly introduced without any qualitative rationale. Our analysis indicates that, in the case of BLJES, our variational distribution (4) can be purely derived from the independence assumption without manually specifying a particular form of a distribution. Although the precise error caused by this independence assumption is still not fully clear, to our knowledge, no information-theoretic BO studies have revealed this way of qualitative justification for the truncate distribution approximation.
> >
> > In Appendix B, we further clarify what happens if 'full independence' of $(\boldsymbol{x}, \boldsymbol{\theta})$ from other points is assumed. Under this stronger assumption, the truncation conditions are removed from the resulting variational distribution as shown in (21). We added an ablation study in Appendix F.8 in which this BLJES without truncation conditions is compared with the original BLJES. The results obviously indicate that BLJES without truncation shows much worse performance than the original BLJES. This is reasonable because BLJES without truncation loses knowledge that $f^{\*}$ and $g^{\*}$ are the maximum values of upper- and lower- problems.

---

### Official Review · Reviewer_kLn9 · 2025-10-30

**Soundness:** 4
**Presentation:** 3
**Contribution:** 4
**Rating:** 6
**Confidence:** 3

**Summary:**

This paper proposes an information-theoretic framework for Bayesian Optimization (BO) in bilevel optimization problems, where both the upper- and lower-level objectives are expensive black-box functions. The authors introduce a new acquisition criterion based on the mutual information (MI) between potential observations and the optimal solutions and values of both levels, termed bilevel information gain. Because the exact evaluation of this MI is intractable, the paper derives a variational lower bound combined with a truncation-based Gaussian approximation to obtain a computationally feasible expression. This approximation extends prior single-level entropy-based BO methods (e.g., Joint Entropy Search) to the bilevel setting. Moreover, the formulation naturally accommodates decoupled observation settings, where upper- and lower-level outputs can be obtained separately, and can be extended to constrained bilevel problems. The proposed method provides the first unified information-theoretic formulation for bilevel BO. The authors present analytical derivations for the lower-bound approximation, demonstrate flexibility under various experimental setups, and validate the approach on Gaussian process sample paths and benchmark test functions. The presentation is mathematically detailed and the empirical results show generally strong performance across tasks. Overall, this work offers a novel and principled direction for handling the intractable mutual information forms arising in bilevel BO and for reducing the computational burden of evaluating complex conditional Gaussian distributions.

**Strengths:**

This paper presents several strengths across originality, quality, and significance.
First, the work introduces a new (MI) gain metric that jointly considers both upper and lower levels in bilevel optimization, addressing a clear gap in existing research. The proposed approach provides a principled way to quantify shared information between levels, supported by a mathematically grounded derivation. By applying a variational lower bound and a truncated Gaussian approximation, the method achieves a computationally tractable form of the otherwise intractable MI expression. Furthermore, the framework is flexible enough to handle decoupled settings, where one level’s observation may be missing, and can also be extended to constrained bilevel problems.
Second, the presentation is clear and well-structured, with solid mathematical explanations and step-by-step reasoning. The authors provide thorough explanations and clarifications for each design choice, helping readers understand the rationale behind the modeling and approximation steps. The logical flow from theoretical formulation to implementation is coherent and easy to follow.
Third, the benchmark results are impressive and superior in most of the tasks. This shows a wide range of applicable cases where fewer drawbacks are seen.
Finally, this work is not only supported by convincing experimental results on benchmark problems but also offers a general mathematical framework that can inspire further research on information-theoretic Bayesian optimization for complex hierarchical problems.

**Weaknesses:**

Despite the solid strengths, there are a few aspects that could be improved.
First, the discussion section does not sufficiently address the limitations of the proposed approach. The paper focuses on the formulation and theoretical derivations but provides little reflection on potential weaknesses or failure cases. The practical use cases are also not deeply explored, leaving readers to infer the applicable scenarios and boundary conditions from the mathematical formulations.


Second, the paper could strengthen its mathematical justification regarding error propagation and convergence.
Third, the presentation is dense with formulas and algebraic derivations but lacks intuitive visual illustrations. A few schematic diagrams or graphical examples would greatly help readers quickly grasp the conceptual flow and the intuition behind the bilevel information gain framework at the beginning.

**Questions:**

There are several questions to help clarify the concerns outlined below:
1. The proposed method heavily relies on approximation techniques, including the variational lower bound and truncated Gaussian assumptions.
Is there any benchmark or ablation study for each of these approximation components?
It would be valuable to include small-scale demonstrations comparing each approximation against ground-truth results or showing quantitative errors.
For instance, while sampling from the Gaussian process posterior distribution $\Omega_B = \{\, y_f(x, \theta),\; y_g(x, \theta),\; f^*,\; g^*,\; x^*,\; \theta^* \,\}$ is indeed challenging, even a simple toy example could help clarify how the random Fourier feature (RFF) approximation behaves and how it affects the estimation of the bilevel mutual information.
2. The above issue may lead to another concern: since multiple approximations are stacked together, compounding approximation errors could occur.
Is there any theoretical assurance of convergence or empirical evidence demonstrating that these cumulative errors remain bounded or do not destabilize the optimization process?
3. In some experimental results, such as Fig. 2 (a), (c) and Fig. 3 (a), the regret curve of BLJES shows a sharp drop within the first 20 iterations, while the BILBO baseline occasionally exhibits similar behavior on other benchmarks.
This appears to be a consistent pattern of the optimizer rather than random variation.
Is there an explanation or intuition for why this phenomenon occurs across certain benchmarks?
I am happy to raise the score if my questions are addressed satisfactorily.
There are further suggestions for improving the presentation:
1. The limitations of this approach should be discussed. This helps the paper show the applicable range of the problem.
2. Visualization illustrations may improve the general ideas. Full of mathematical formulas requires more effort from viewers.

---

> ### Author Response · Authors · 2025-12-03
>
> Thank you for your constructive suggestions. We summarize our answer as the following five subjects (reference can be found in our paper).
>
> 1.
>
> > practical use case
>
> See response #1 to reviewer pjE9.
>
> 2.
>
> > approximations
>
> The major approximation components are the lower bound, RFF, and the MC sampling. For the lower bound, although directly revealing the tightness is still challenging. Instead, we newly analyzed when the lower bound becomes tight. We derived that our variational distribution is equal to the true distribution if the conditional independence assumption holds, for which details are in Appendix B. For RFF, a new ablation study is shown in Appendix F.9, in which performance of the original GP based sampling and RFF is compared by using the BG benchmark function. Although we currently do not have a theoretical analysis of the stacked error, we newly added the empirical evaluation of the approximation error caused by the combination of RFF and the MC sampling (Appendix F.10). Overall, the error analysis is still a challenging issue for information-theoretic BO methods even in the standard single level setting, but our empirical evaluations throughout the entire experimental sections suggest that the approximated acquisition function of BLJES reasonably worked in practice.
>
> 3.
>
> > sharp drop
>
> We are not fully sure which behavior the reviewer indicated, but we guess that it might be the issue of the log-scale plots. In the log-scale plot such as Fig 3 in the revised manuscript, the lower edge of the interval plot is depicted as longer than the upper edge, because of asymmetry of the vertical axis. Particularly when the lower edge of the interval becomes less than 0, the shaded region can be suddenly dropped. Since the optimizers (BLJES and BILBO) have more chances that the lower edge becomes less than 0, this can happen more than random selection.
>
> 4.
>
> > limitation
>
> We added section 6 for limitations, in which we discuss about issues on the dimensionality and theoretical analysis.
>
> 5.
>
> > illustration
>
> We added a schematic illustration of random variables in our MI at Fig. 2.

---

### Official Review · Reviewer_pjE9 · 2025-10-31

**Soundness:** 3
**Presentation:** 3
**Contribution:** 2
**Rating:** 4
**Confidence:** 4

**Summary:**

This paper proposes to study an open problem in bi-level optimisation that both the upper level and the lower level involve expensive black-box functions. The authors design a new and interesting method to solve it. They theoretically and empirically verify the proposed method.

**Strengths:**

The problem is well defined. The theory looks solid and rigorous, though I did not get time to check all details in the proofs.

**Weaknesses:**

The proposed problem is not well motivated. I would like to ask:

- What applications are aligned with the setting that both the upper level and the lower level involve expensive black-box functions?
- Do existing methods work in this setting? If no, why? What would happen if they are applied here?

The experiments are not strong enough. Only one relatively small real dataset is used - more datasets are required. For ablation study, the authors only present results on number of samplings. No code is provided.

**Questions:**

Please see above.

---

> ### Author Response · Authors · 2025-12-03
>
> Thank you for your constructive comments. We summarize our answer as the following five subjects (reference can be found in our paper).
>
> 1.
>
> > applications
>
> As described in introduction, bilevel optimization has been studied in a variety of contexts such as optimal control, chemical reaction optimization, and shape optimization. Most of those existing studies focus only on the case that (at least) a lower level problem is not expensive. On the other hand, for example, consider the case that the lower level problem is defined by a simulator of a subject of interest. If the computation of the simulator is expensive, it immediately becomes a bilevel problem in which both level observations are expensive.
>
> We consider that this scenario is particularly important for scientific applications such as simulator based explorations of inorganic crystals. A typical setting is that the upper level problem is the optimization of some physical property (such as ion conductivity of battery materials) and the lower level problem is the energy minimization (stable structure search). If the simulator is the quantum-mechanical calculation, which is widely used in a variety of scientific fields, the observation cost can be high (e.g., can be more than a day to obtain one observation depending on the model size). We believe that this topic will become increasingly important with the growing attention to AI for Science across a wide range of scientific fields.
>
> Related descriptions were added at the end of the second paragraph of introduction.
>
> 2.
>
> > Do existing methods work in this setting?
>
> Most of existing methods assume an inexpensive lower-level problem and therefore they do not seriously care about the observation cost of the lower problem. For example, Wang et al., IEEE CEC 2021 consider an algorithm in which 60 times lower-level observations are performed for `each of' 100 times upper-level observations. This means that the total observations are 60x100 = 6,000 times lower-level observations and 100 times upper-level observations.
>
> Our experiments performed 100 iterations, in each of which one upper and one lower observations were obtained. In other words, our setting has the total 200 observation budget including both the upper- and the lower- level observations. If the above existing setting is applied to our experiment, only the four times observation of the upper-level problem exceeds the budget (60x4 + 4 > 200). Therefore, we focus on comparison with a method (BILBO) assuming high evaluation cost for both level observations.
>
> 3.
>
> > experiments are not strong. Only one relatively small real dataset
>
> We added real dataset called the Chemical dataset (see section 5.2).
>
> As we mentioned in the second and the third paragraphs of introduction, (both-level expensive) bilevel BO has been studied only by a few papers. Therefore, unlike the standard problem setting of BO, there is not a rich supply of real-world datasets for our setting from existing studies. A seminal paper of bilevel BO (Chew et al. ICML 2025) introduced two real datasets, both of which we already used, and in their paper, performance on four benchmark functions and two real datasets were reported. In our main text, we reported nine settings of GP derived synthetic functions, five benchmark functions, and two real-datasets.
>
> Further, in appendix, we showed results on a stronger noise setting (Appendix F.2), sensitivity to the number of samplings (Appendix F.4), the continuous domain setting (Appendix F.5), the extension for the constraint problem (Appendix F.6), higher dimensional settings (Appendix F.7, newly added and described below), comparison with a simplified variant of MLJES (Appendix F.8, newly added and related to response #4 to reviewer k4Ay), the effect of RFF approximation (Appendix F.9, newly added and related to response #2 to reviewer kLn9), and the combined error of the MC sampling and RFF (Appendix F.10, newly added and related to response #2 to reviewer kLn9).
>
> If 'small' indicates the input dimensions, see our response #2 to reviewer k4Ay.

---

> > ### Author Response · Authors · 2025-12-03
> >
> > 4.
> >
> > > ablation
> >
> > We added two ablation studies.
> >
> > First, in Appendix F.8, we performed comparison with a simplified variant of BLJES. In this variant, the truncation conditions $f(\boldsymbol{x}, \boldsymbol{\theta}^{\*}(\boldsymbol{x})) \leq f^{\*}$ and $g(\boldsymbol{x}^{\*}, \boldsymbol{\theta}) \leq g^{\*}$ are removed from (5), which results in a simplified lower bound (shown in (21)). The purpose here is to evaluate importance of these truncation conditions (BLJES without truncation corresponds to imposing a strong independence assumption described in detail at Appendix B). The results in Appendix F.8 obviously indicate importance of the truncation conditions.
> >
> > Second, in Appendix F.9, we added comparison between BLJES with the RFF-based sampling (which we employ in the main experiments as described in section 3.2) and with the original GP posterior sampling. We require the RFF sampling to obtain samples (such as $\tilde{x}^{\*}$) from continuous space. On the other hand, in the pool setting, we can compare performance of these two sampling strategies. In the results, we do not see a large difference between two sampling strategies, which suggests RFF did not largely deteriorate the performance.
> >
> > 5.
> >
> > > code
> >
> > We uploaded our code as supplementary.

---

### Meta-Review · Area_Chair_ej8H · 2026-01-07

**Summary:**

This paper proposed an information-theoretic Bayesian optimization framework for black-box bilevel optimization, where both the upper- and lower-level objectives are expensive black-box functions. It proposed an information-theoretic approach for bilevel optimization, which considers the information gain of both the upper- and lower-optimal solutions and values. Moreover, it provided analytical derivations for the lower-bound approximation, demonstrate flexibility under various experimental setups, and validate the approach on Gaussian process sample paths and benchmark test functions.  It also provided some numerical experiments to demonstrate effectiveness of the proposed method.

This paper studied the black-box bilevel optimization, but many related methods such as zeroth-order methods [1-5] have not been mentioned. Meanwhile, in the experiments, comparisons are insufficient. The authors should add some zeroth-order methods for black-box bilevel optimization as comparisons.

[1] Fully Zeroth-Order Bilevel Programming via Gaussian Smoothing

[2] Optimal Zeroth-Order Bilevel Optimization

[3] Provable Faster Zeroth-order Method for Bilevel Optimization with Optimal Dependency on Error and Dimension

[4] Stochastic Regret Guarantees for Online Zeroth- and First-Order Bilevel Optimization

[5] DTZO: Distributed Trilevel Zeroth Order Learning with Provable Non-Asymptotic Convergence

**Reviewer Concerns:**

Although the authors have provided rebuttals to address some reviewers' concerns, there exist some concerns are addressed. For example, Reviewer pjE9 think that  this paper do not provide some specific application cases in AI and machine learning.

**Reviewer Scores:**

Although the authors have provided rebuttals to address some reviewers' concerns, there exist some concerns still are not addressed. Thus, the likelihood of drastically changing the score is low.

---

### Decision · Program_Chairs · 2026-01-26

Reject